# Coupling and Coordination between Tourism, the Environment and Carbon Emissions in the Tibetan Plateau

**Jiayuan Wang [1], Lin Yi [1,*], Lingling Chen [2,*], Yanbing Hou [3], Qi Zhang [1] and Xuming Yang [1]**

[1] College of Geography and Planning, Chengdu University of Technology, Chengdu 610059, China; circlewang29@gmail.com (J.W.); zq1008b@gmail.com (Q.Z.); translatorchrisy@163.com (X.Y.)
[2] Department of Tourism Management, Jinling Institute of Technology, Nanjing 211169, China
[3] School of Mathematics, Sichuan University, Chengdu 610041, China; hyolandab531@163.com
* Correspondence: yilin57@163.com (L.Y.); chenlingling@jit.edu.cn (L.C.)

**Abstract:** Studying the relationships among tourism, the environment and carbon emissions is key to understanding how tourism activity affects the sustainable development of tourism in the Tibetan Plateau. Using Lhasa, Tibet, as a case study, the coupling and coordination relationships among the three systems were analysed to explore the impact of tourism behaviour on sustainable tourism development. Utilising panel data from 2010 to 2020, the carbon emissions of tourism activities were calculated using a bottom-up approach. The coupling coordination model was employed to examine the interrelationship of the economy, the ecological environment and carbon emissions. Additionally, the Tapio model was utilised to further analyse the dependency relationship between economic development and carbon emissions. This assessment of the sustainability of Lhasa's tourism industry revealed that (1) the economy, the environment and carbon emissions are indeed closely intertwined with sustainable development and that (2) there has been a significant increase in the coupling coordination among the economy, the environment and carbon emissions from 2010 to 2020, coupled with a gradual decrease in economic dependency on carbon emissions. Despite providing favourable conditions for sustainable development, there remains considerable disparity among the three subsystems, with relatively low overall coordination. Accordingly, some practical low-carbon tourism policies are recommended to guide tourism practices and promote long-term sustainability.

**Keywords:** Tibetan Plateau; tourism industry; carbon emissions; Tapio model; coupling coordination; sustainable development

## 1. Introduction

Tourists flock to the Tibetan Plateau for its stunning natural beauty, unique culture and serene atmosphere [1]. Tourism is a key economic driver that is fuelling rapid economic development in the region [2]. However, tourism in the Tibetan Plateau is still in its early stages of development [3]. It relies significantly on natural resources and the surrounding environment [4]; however, as an important indicator of global climate change [5], the ecological environment of the Tibetan Plateau is particularly and irreversibly susceptible to human activity owing to its high altitude [6], dry climate [7] and limited carrying capacity [8]. Unreasonable resource development and excessive tourism activity have increased environmental pressures [9]. If these are not controlled, the fragile environment of the Tibetan Plateau will be increasingly degraded, especially under gradually worsening global climate change [10–12]. Therefore, there is an urgent need to determine whether a balance between tourism activities and preserving the ecological environment is possible.

Existing studies on the region have predominantly focused on permafrost responses [13], ecological research [14] and rangeland degradation [15]. Government departments have managed and protected the environment in many ways, such as by identifying the ecological baselines, implementing ecological projects, developing mineral resources and limiting highly polluting enterprises [16]. However, the impacts of tourism have received

little study [17]. Tourism activity can promote regional economic development, but it also puts pressure on the ecological environment through carbon emissions [18,19]. When environmental degradation exceeds a threshold, the cost of economic activity is increased. This places heavy constraints on the sustainable growth of the regional tourism sector [20]. Therefore, studying the relationships among economic growth, carbon emissions and the ecological environment is essential. Achieving a balance between tourism development and environmental protection is key to sustainable tourism.

From as early as 1920, studies have focused on the relationships between tourism and the natural environment [21]. These have included studies on tourism carbon emissions (TCEs) [4,22], carbon emissions efficiency [23], and coupling and coordination relationship (CCRs), reflecting the coordinated development level between economic development (ED) and the ecological environment (EE). It has been proven that the growth of ED increases TCEs [24,25]. Several CCRs of the tourism–economy–environment system have been reported [26,27]. However, these studies ignored the quantitative analysis of the impact of tourism behaviours on the ecological environment and tourism economic development [28–30]. They mainly focused on CCRs between ED and EE in areas with advanced tourism development, such as the Yangtze River Economic Belt, Northeast China, Finland and Kyushu, Japan. Few studies have focused on the CCRs among the TCEs, ED and EE of the plateaus [31]. Thus, it is difficult to explain how different tourism behaviours affect the sustainable development of tourism and to implement practical interventions to guarantee the sustainable development of tourism.

Additionally, this study focused on the influence of tourism behaviours on the sustainable development of tourism in the Tibetan Plateau. This study combined the Tapio decoupling model and the coupling coordination degree model to analyse the CCRs between TCEs, ED and EE. This more accurately revealed the influence relationship and intensity of tourism behaviours from multiple dimensions. The results will also help in the formulation of practical policies to influence tourism behaviours and thereby promote the long-term sustainable development of tourism.

## 2. Mechanism of Sustainable Tourism Development

The Tibetan Plateau attracts many tourists with its unique natural landscapes [32], which is the fundamental basis for tourism development [33]. The activities of these tourists yield positive returns for the tourism economy [34], boosting the growth of industries like hotels, restaurants and retail [35]. However, these activities also produce carbon emissions and thus have environmental impacts [36]. Without proper attention to environmental protection, the degradation of the environment will diminish the allure of the plateau's natural landscapes. Consequently, there will be a decrease in tourism activities, leading to a reduction in ED as well. The carbon emissions from tourism activities can be seen as a hub between the tourism economy and the environment and could be used to accurately quantify the relationship between economic growth and environmental protection (as shown in Figure 1).

Today, environmental protection is a top priority for nations worldwide and sacrificing the environment is not considered an acceptable cost of ED. Rather, reducing TCEs is viewed as an effective strategy for balancing energy consumption and mitigating environmental degradation [37,38]. Consequently, governments are investing substantial funds in promoting the benefits of low-carbon tourism and encouraging tourists to engage in environmentally friendly travel practices [39]. Tourism economic growth is an important financial backbone for implementing some concrete measures to reduce TCEs in the plateau [17,40]. The interplay among TCEs, EE and ED mutually reinforces and constrains each factor; these three components are indispensable constituents of the tourism industry, playing a crucial role in promoting sustainable development within the tourism sector.

With the acceleration of ED, many regions have begun to focus on whether the tourism industry can achieve sustainable growth. For example, in areas with good EE and low TCEs, green and low-carbon standards have been achieved [41]. In areas with prosperous

ED and a good EE, harmonious environmental protection has progressed. In economically prosperous regions where TCEs are up to standard, there has been notable technological and industrial development [42]. Sustainable tourism development can be defined as the development of a tourism economy that does not destroy the local natural environment, making rational use of tourism resources and protecting the developed resources [35]. This sustainable development can only be achieved through the mutual coordination of TCEs, EE and ED [20].

Accordingly, this study focused on the three aspects of tourism that contribute the most carbon emissions: transportation, accommodation and activities. These aspects were chosen to represent the overall carbon footprint of tourism.

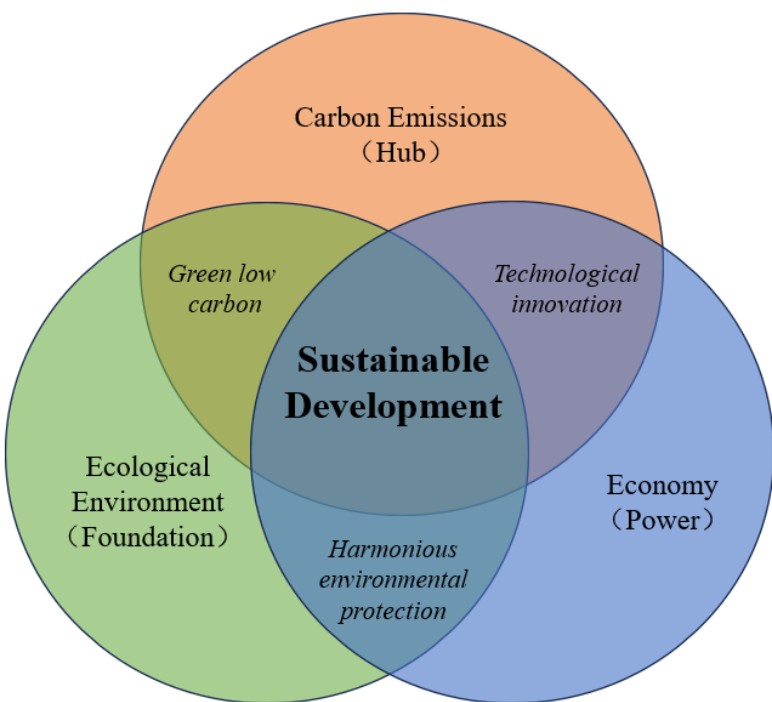

**Figure 1.** Issues in sustainable tourism development.

## 3. Materials and Methods

### 3.1. Study Area

Lhasa is situated at the centre of the Tibetan Plateau (29°36′ N, 91°06′ E; Figure 2), covering an area of 29,655.5 km². The plateau's average altitude of 3650 m is relatively high, making it known as the world's 'third pole'. It is the provincial capital and economic cultural centre of the Tibet Autonomous Region. One of the tributaries of the Yarlung Zangbo River flows through Lhasa. This area not only has rich historical significance but also features scenic spots, making it abundant in tourism resources [43]. Within Lhasa, Potala Palace, Jokhang Temple and Norbulingka are listed as cultural World Heritage Sites.

Over the past few years, there has been a notable surge in tourism growth in Tibet. The total tourism revenue in Lhasa increased by CNY 25.973 billion from 2010 to 2020, which is equivalent to 38.30% of Lhasa's 2020 GDP [44]. As a proportion of GDP, tourism revenue increased from 23.50% to 44.50% over this period. However, the Tibetan Plateau has low precipitation, poor geographical conditions and a fragile ecology [45]. This makes the relationship between tourism and the environment highly important and its study essential.

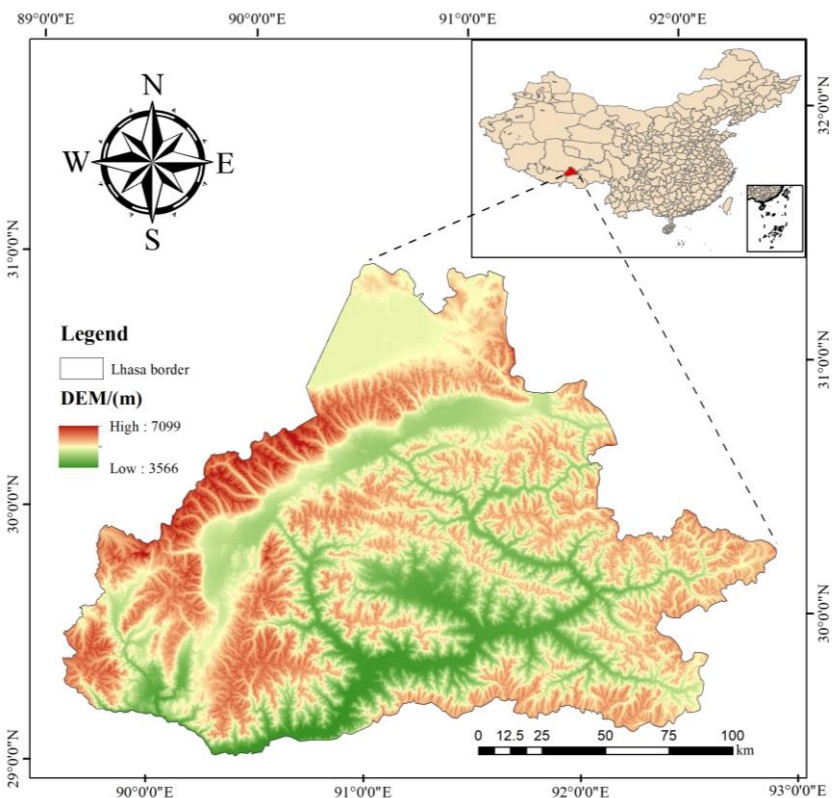

**Figure 2.** Topographical map of the Lhasa study area and its location in China (inset).

### *3.2. Data Collection*

The data used in this study included information on TCEs, ED and EE. Data on *TCEs, carbon productivity*, *TCE intensity*, *GDP*, *Chinese tourist numbers* and *per capita tourism consumption in Lhasa* were obtained from *The Yearbook of China Statistics* [44], China tourism statistics [44], Qinghai statistics [46], China city statistics [44], Tibet statistics [47] and tourism sampling survey data [48]. Data on *sewage drainage*, the *forest coverage ratio* and the *percentage of forest cover* were obtained from *The Yearbook of China Environmental Statistics* [49], the China Statistical Annual Report of Ecological Environment [49] and the official website of the Lhasa Tourism Bureau for the period 2010–2020. Specific data and sources are shown in Table 1.

**Table 1.** Data sources.

| Data | Source |
| --- | --- |
| TCEs, number of Chinese tourists | *The Yearbook of China Tourism Statistics* |
| Carbon productivity, TCE intensity | *The Yearbook of Qinghai Statistics and Tibet Statistics* |
| GDP | *The Yearbook of China Statistics* |
| Per capita tourism consumption in Lhasa | *Tourism Sampling Survey Data, The Yearbook of Qinghai Statistics and China Tourism Statistics* |
| Sewage drainage | *The Yearbook of China Environmental Statistics*; *China Statistical Annual Report of Ecological Environment* |
| Percentage forest cover | *China Statistical Annual Report of Ecological Environment*, the official website of the Lhasa Tourism Bureau (https://lfj.lasa.gov.cn/lsslyfzj/zwgk/zwgk.shtml, accessed on 20 January 2023) |

### 3.3. Estimation of TCEs

TCEs were estimated by combining data on tourism traffic, tourism accommodation and tourism activity [50]. There is no independent system accounting for industrial carbon emissions and establishing tourism satellite accounts in China's tourism industry [51]. Therefore, a top-down approach to calculating TCEs could not be applied. A bottom-up method [52] based on human activity is more time-sensitive and flexible than a top-down method [53]. Hence, the TCEs were computed using the bottom-up method, as shown in Equation (1):

$$C_Y = C_{YT} + C_{YH} + C_{YE} \tag{1}$$

Here, $C_Y$ denotes the total TCEs in year $Y$ and $C_{YT}$, $C_{YH}$ and $C_{YE}$ denote the traffic TCEs, accommodation TCEs and activity TCEs in year $Y$, respectively.

The specific calculations for each type of TCE are as follows.

#### 3.3.1. Estimation of Traffic TCEs

According to previous studies, energy consumption can be used as a proxy for carbon emissions [25]. The energy consumption of tourism traffic was calculated according to Equation (2):

$$C_{YT} = \sum_{j=1}^{n} \vartheta \times M_j \times B_j \times Q_{Tj} \tag{2}$$

Here, $C_{YT}$ refers to the transport TCEs, $\vartheta$ refers to the proportion of passengers who are tourists, $M_j$ denotes the total count of passengers opting for transport mode $j$ (aircraft, road vehicles or trains), $B_j$ refers to the travel distance using transport mode $j$ and $Q_{Tj}$ stands for the energy consumption coefficient of transport mode $j$ (MJ·pkm$^{-1}$). The values for the train, road vehicle and aircraft traffic modes were set to 1.0, 1.8 and 2.0 g MJ·pkm$^{-1}$, respectively [54–57].

#### 3.3.2. Estimation of Accommodation TCEs

There are many hotels and homestays in Lhasa. Taking data availability into account, this work computed the carbon emissions of star-rated hotels as a proxy for accommodation TCEs, as shown in Equation (3) [25]:

$$C_{YH} = 365 \times C_Y \times Q_Y \times R \times S \times \frac{1}{1000} \times \frac{44}{12} \tag{3}$$

Here, $C_{YH}$ signifies accommodation TCEs, $C_Y$ refers to the number of beds in star-rated hotels in year $Y$, $Q_Y$ refers to the rate of guest room tenancy in year $Y$, $R$ signifies the energy consumption factor of star-rated hotels per bed-night, $S$ refers to the $CO_2$ emission factor per bed-night, 1/1000 is a unit conversion coefficient and 44/12 refers to the ratio of the molecular weight of $CO_2$ to the atomic weight of $Q_Y$. Based on existing studies [52,58] and the operating status of star-rated hotels, the energy consumption coefficient was determined as 155 MJ·bed$^{-1}$. The conversion between energy consumption and TCEs is 43.2 g $CO_2$·MJ$^{-1}$.

#### 3.3.3. Estimation of Activities TCEs

According to the literature, tourism activities can be categorised into *sightseeing*, *leisure*, *business*, *family visits* and *other*. Their TCEs can be estimated using Equation (4):

$$C_{YE} = \sum_{i=1}^{n} Q_i \times H_i \times \tau_i \tag{4}$$

Here, $Q_i$ and $H_i$ refer to the total tourist number and the ratios involved in different activities, respectively, and $\tau_i$ refers to the TCE coefficient of tourism activity $i$. According to existing studies [58], the TCE coefficients are as follows: *sightseeing* = 417 g·person$^{-1}$,

*leisure* = 1670 g·person$^{-1}$, *business* = 786 g·person$^{-1}$, *family visits* = 591 g·person$^{-1}$ and *other* = 172 g·person$^{-1}$.

*3.4. Model of Sustainable Tourism Development*

3.4.1. Model of Coupling Coordination Degree

The *coupling degree* (*C*) only reflects the degree of interaction between systems and cannot accurately depict the actual level of synergy [28]. *The coupling coordination degree* (*D*) refers to the degree of alignment between subsystems during a specific period, indicating the quality of coordination [59]. Higher values suggest the relatively balanced development of each subsystem. Therefore, by measuring *D*, the interaction strength between TCEs, ED and EE could be assessed, which would enable the degree to which the tourism industry in the plateau is coordinated with the environment to be analysed [17]. The model was constructed based on the degree of coupling and coordination among the three subsystems, as shown in Equation (5):

$$D = \sqrt{T \times C} \tag{5}$$

Here, *D* refers to the *coupling coordination degree*, *T* refers to the comprehensive coordination index of subsystems and *C* refers to the *coupling degree* of subsystems.

1. The index *T* was calculated using Equation (6):

$$T = \delta F_1 + \varepsilon F_2 + \theta F_3 \tag{6}$$

Here, $\delta$, $\beta$ and $\gamma$ refer to the coefficients of $F_1$, $F_2$ and $F_3$, respectively. Given the equal importance of TCEs, ED and EE, each contributes 1/3 to the overall assessment. $F_1$, $F_2$ and $F_3$ represent the coordination evaluation values of TCEs, ED and EE, respectively. Their values were calculated according to Equation (7) [23]:

$$F_1 = \sum_{i=1}^{m} a_i x_i, F_2 = \sum_{i=1}^{n} b_i y_i, \ F_3 = \sum_{i=1}^{k} c_i z_i \tag{7}$$

Here, $x_i$, $y_i$ and $z_i$ are the coordination evaluation indices of traffic, the tourism industry and the natural environment, respectively, after dimensionless treatment, and $a_i$, $b_i$ and $c_i$ are their weights. The selection of indices was determined by sustainable development connotation, ecological civilisation construction requirements, green development theory and their interaction mechanisms (Table 2). Eight tertiary-grade indices involving the amount of TCEs and their status were used. The ED was estimated by 11 indices involving three aspects: tourism market scale, tourism industry level and tourism growth rate. The EE contained four indices covering three aspects: environmental pollution, environmental investment and governance. Therefore, in Equation (7), $m = 8$, $n = 11$ and $k = 4$. The indices were divided into positive and negative categories according to their contributions to sustainable tourism development [60]. The data collected in this paper passed tests of reliability and validity. The original data were pre-processed via max–min standardisation [61].

The entropy weight method is a weight determination method based on information entropy. Because this method is not limited by the type and distribution of indices, it is suitable for various types of data and problems. It also takes into account the uncertainty and randomness of data and can objectively evaluate the importance of each index [62]. Therefore, the entropy weight method was selected to calculate the index weight.

**Table 2.** Indices used for evaluating the coordination between TCEs, ED and EE.

| Primary | Secondary | Tertiary | Attribute | Weight |
|---|---|---|---|---|
| TCEs | Amount of TCEs | *Traffic TCEs* (t) (A11) | - | 0.14 |
| | | *Accommodation TCEs* (t) (A12) | - | 0.08 |
| | | *Activity TCEs* (t) (A13) | - | 0.09 |
| | | *Total TCEs* (t) (A14) | - | 0.14 |
| | TCE status quo | *TCE intensity* (t/CNY hundred million) (A21) | - | 0.12 |
| | | *TCE density* (t/km$^2$) (A22) | - | 0.14 |
| | | *Carbon productivity* (hundred million/t) (A23) | + | 0.18 |
| | | *Per capita TCEs of tourists* (t/million) (A24) | - | 0.12 |
| ED | Tourism market scale | *Tourists number in Lhasa* (millions) (B11) | + | 0.10 |
| | | *Lhasa GDP* (CNY hundred million) (B12) | + | 0.11 |
| | | *ED of Lhasa* (CNY hundred million) (B13) | + | 0.11 |
| | | *Domestic tourism revenue* (CNY hundred million) (B14) | + | 0.11 |
| | | *Domestic tourist number* (hundred million) (B15) | + | 0.09 |
| | | *Tourism consumption per person* (CNY million) (B16) | + | 0.11 |
| | | *Total tertiary industry* (CNY hundred million) (B17) | + | 0.10 |
| | Tourism industry level | *Total ED as a proportion of tertiary industry* (B21) | + | 0.10 |
| | | *Total ED as a proportion of GDP* (B22) | + | 0.10 |
| | Tourism growth rate | *Increase rate of ED* (B31) | + | 0.03 |
| | | *Growth rate of tourist number* (B32) | + | 0.03 |
| EE | Environmental pollution | *SO$_2$ emissions* (t) (C11) | - | 0.21 |
| | | *Discharge of sewage* (ten thousand t) (C12) | - | 0.21 |
| | Environmental investment and governance | *Nature reserves number* (C21) | + | 0.24 |
| | | *Percentage of forest cover* (C22) | + | 0.35 |

First, the relative entropy of each index was calculated according to Equations (8) and (9):

$$T = -\ln(y)^{-1}\sum_{i=1}^{n} Q_{ij}\ln Q_{ij} \tag{8}$$

$$P_{ij} = \frac{M_{ij}}{\sum\limits_{i=1}^{n} M_{ij}} \tag{9}$$

Here, $T_j$ refers to the relative entropy of each index, $y$ refers to the year ($y = 10$), $M_{ij}$ refers to the standardised value of the $j$th indicator in category $i$, and $Q_{ij}$ refers to the proportion of the value of parameter $M_{ij}$ in each year to the total value.

Then, the weight of each index was determined according to Equation (10):

$$w_j = \frac{1 - T_j}{a - \sum\limits_{i=1}^{a} T_j} \tag{10}$$

Here, $w_j$ refers to the weight of each index and $a$ refers to the number of indices.

2.  Calculation of $C$:

The degree of coordination between the three subsystems ranges between 0 and 1 and is denoted by $C$. When $C$ approaches 1, the coordination of subsystems is relatively strong and proceeds in an ordered direction. Conversely, the coordination among sub-systems is relatively weak and leads to disorder when $C$ approaches 0.

$$C = \left\{ \frac{F_1 \times F_2 \times F_3}{\left[\frac{F_1 + F_2 + F_3}{3}\right]^3} \right\}^{\frac{1}{3}} \tag{11}$$

3.  Measurement of $C$ and $D$.

4. Inspired by the classification of *D* between systems reported by Liu [63] and Qi [64], the classification criteria were determined based on the actual situation of the plateau, as shown in Tables 3 and 4.

**Table 3.** Ranges of system coupling degree *C* and their corresponding phases.

| C | Phase |
|---|---|
| $0 \leq C \leq 0.3$ | Low-level coupling |
| $0.3 < C \leq 0.5$ | Antagonism phase |
| $0.5 < C \leq 0.8$ | Running-in stage |
| $0.8 < C \leq 1$ | High-level coupling |

**Table 4.** Ranges of *D* and their corresponding categories.

| D | Category | Sub-Category |
|---|---|---|
| $0 < D \leq 0.3$ | Low coordination | Disorder |
| $0.3 < D \leq 0.5$ | | Reluctant harmony |
| $0.5 < D \leq 0.7$ | Moderate coordination | Primary harmony |
| $0.7 < D \leq 0.8$ | | Intermediate harmony |
| $0.8 < D \leq 1.0$ | High coordination | Harmony |

### 3.4.2. Calculation of the Tapio Decoupling Index

*Decoupling* refers to the relationship between the various parts of a system [65]. The decoupling theory intuitively reflects the dependence of ED on TCEs. The Tapio decoupling model is widely used across various fields due to its high stability and effectiveness in studying interconnected relationships among elements [66]. The decoupling of TCEs and ED was analysed using the Tapio decoupling model, as shown in Equation (12):

$$e = \frac{(\Delta T / T)}{(\Delta H / H)} \tag{12}$$

Here, *e* represents the Tapio decoupling index (Tapio index) and $\Delta T$ and $\Delta H$ represent the amplitudes of variation in TCEs and ED, respectively. The size of the tourism economy was estimated from its total revenue. Following Tapio's [66] classification of the decoupling index, the classification criteria were established, as shown in Table 5.

**Table 5.** Decoupling states.

| Type | $\Delta T$ | $\Delta H$ | e | Decoupling State |
|---|---|---|---|---|
| Negative decoupling | >0 | >0 | >1.2 | Very strong negative decoupling |
| | >0 | <0 | <0 | Strong negative decoupling |
| | <0 | <0 | $0 < e < 0.8$ | Weak negative decoupling |
| Decoupling | >0 | >0 | $0 < e < 0.8$ | Weak decoupling |
| | <0 | >0 | <0 | Strong decoupling |
| | <0 | <0 | >1.2 | Recessive decoupling |
| Link | >0 | >0 | $0.8 < e < 1.2$ | Expansive coupling |
| | <0 | <0 | $0.8 < e < 1.2$ | Recessive coupling |

### 3.4.3. Geographical Detector Method

The geographical detector method can objectively reflect the influence of an independent variable on the dependent variable [67]. It was used to verify the impact of factors on the coordinated development of tourism in this study, as shown in Equation (13).

$$q = 1 - \frac{\sum_{r=1}^{s} K_r \rho_r^2}{K \rho^2} \tag{13}$$

Here, $q$ refers to the degree of influence of a factor on $D$, where the closer the q value is to 1, the greater the influence of the factor on $D$; $s$ refers to the number of influencing factors ($s = 7$); $K$ refers to the number of samples across the entire region ($K = 10$); $K_r$ refers to the number of samples in each layer; $\rho^2$ refers to the overall variability of $D$ across the entire region; and $\rho_r^2$ refers to the variance in $D$ in each layer.

## 4. Results

### 4.1. Changes in TCEs in Lhasa (2010–2020)

Using Equations (1)–(4) and the bottom-up estimation approach, the total TCEs in Lhasa were calculated for the period spanning 2010 to 2020. The results are shown in Figure 3. The total TCEs in Lhasa showed an upward trend followed by a downward trend. According to these trends, tourism development in Lhasa was divided into three stages: gradual (2010–2015), rapid (2015–2019) and declining (2019–2020).

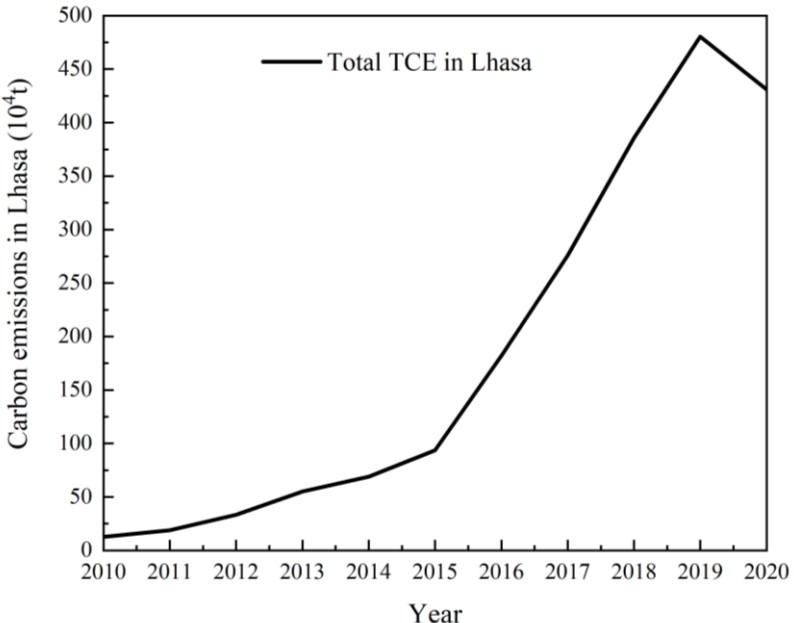

**Figure 3.** Tourism-related TCEs in Lhasa, 2010–2020.

In the gradual period, the total TCEs increased steadily at an average growth rate of 49.04%. The TCEs reached 0.94 million tons in 2015, which marked a turning point and the beginning of the rapid growth stage. The cumulative TCEs then surged from 0.94 million tons to 4.80 million tons within a mere four years. This substantial increase of 3.87 million tons occurred at an average growth rate of 50.49%.

In 2020, the TCEs decreased for the first time in 10 years, by 49.60 million tons. The growth rate declined sharply by −10.33%, which was primarily attributed to significantly reduced tourist numbers during the COVID-19 pandemic. This downturn led to corresponding decreases in TCEs from human activities.

### 4.2. Contribution of Components to TCEs

Notably, the trend in A11 (as defined in Table 2, the following indices are also referred to) closely paralleled that in A14. Among the three categories of tourism activities, traffic was the predominant contributor to TCEs, generating a significantly higher amount than other categories (accommodation and tourism activities). Contributions of the most important travel modes (Figure 4)—aircraft, road vehicles and trains—are mainly analysed here.

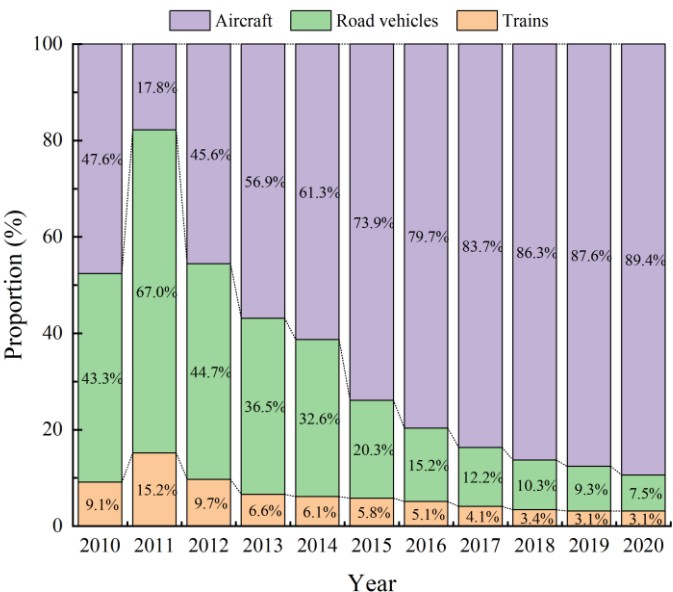

**Figure 4.** Chronological trends in proportions of road vehicle, train and aircraft TCEs.

The proportion of TCEs from trains and road vehicles generally declined but the total TCEs increased. The TCEs from trains increased substantially, climbing from 0.094 million tons in 2010 to 1.325 million tons in 2020, with an average annual growth rate of 30.3%. Road vehicle TCEs also increased from 0.448 million tons in 2010 to 3.189 million tons in 2020, reflecting an annual average growth rate of 21.7%. The total amount of TCEs from aircraft was the highest and showed the largest increase, surging from 49.17 million tons in 2010 to 3814.44 million tons in 2020, an average annual growth rate of 54.5%. The TCEs per kilometre of trains, road vehicles and aircraft travel were 65, 132 and 396 g, respectively [68]. Aircraft TCEs were six-fold higher than train TCEs and three-fold greater than road vehicle TCEs per kilometre. Due to their convenience, aircraft are strongly preferred by tourists, and their use has increased in the last decade [69]; however, they are expensive and generate the most TCEs.

A comparative analysis suggested that road vehicle travel has high flexibility and allows free travel to scenic spots. Trains have high capacity, low cost and generate the least TCEs [58]. Government departments should promote tourism by trains or road vehicles and develop low-carbon fuels (biofuels or hybrid power sources) to reduce traffic TCEs and promote sustainable tourism development.

### 4.3. Variations in Indices Making Positive and Negative Contributions to Tourism Sustainability (2010–2020)

By analysing the changes in each index, the impact of the three subsystems—TCEs, ED and EE—on the progress of sustainable tourism development could be discerned. The indices were computed utilising Formula (7), and the outcomes are visually presented in Figure 5.

TCE indices generally showed upward trends, which indicated that TCEs in Lhasa increased during 2010–2020. Most indices first increased and then decreased, while A24 increased from 2010 to 2020. This may have been due to the opening of Lhasa–Chongqing–Shanghai Airlines in 2012 [70]. This spurred domestic tourism from inland and coastal regions to Lhasa, rapidly boosting TCEs. From 2015 to 2019, the growth rates of all indices increased. This was because of some policies issued by the Lhasa Municipal Tourism Bureau in 2015 to strengthen the tourism infrastructure and expand tourism development [71]. After 2019, almost all TCE indices decreased rapidly, which was due to the COVID-19 outbreak.

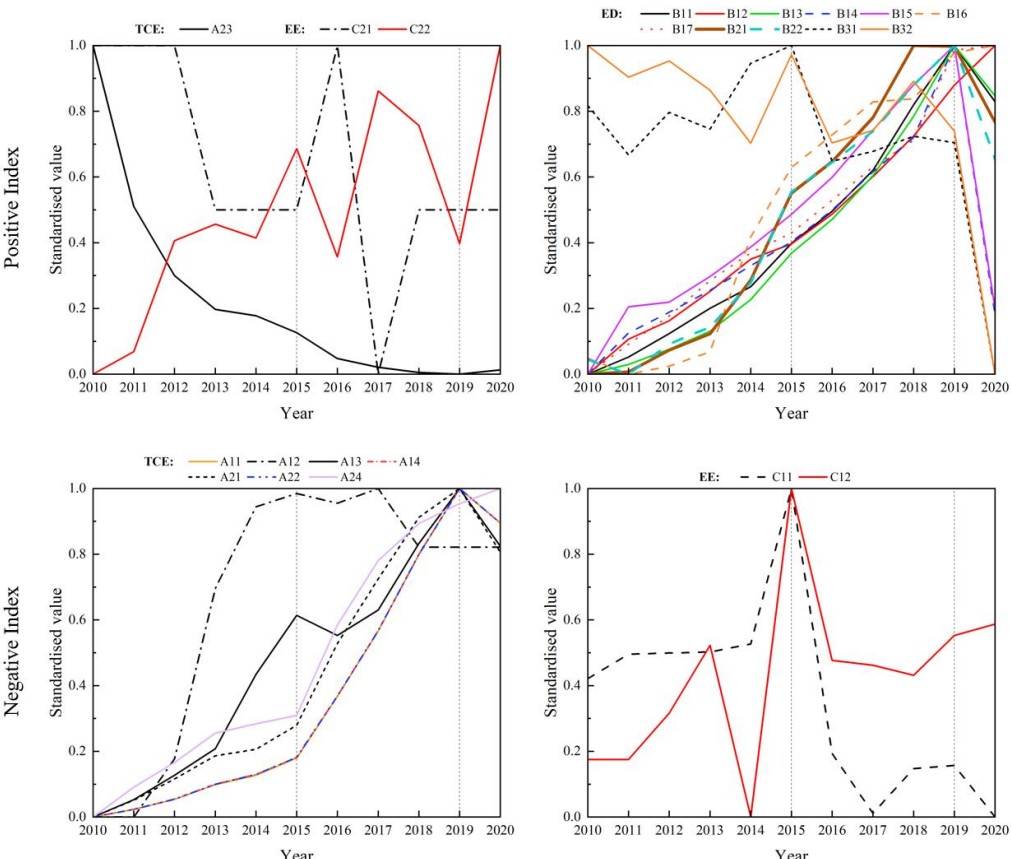

**Figure 5.** Variations in positive and negative tourism indices in Lhasa, 2010–2020.

Regarding the negative indices of EE, 2015 was an inflection point, with C11 and C12 falling sharply after 2016. In terms of positive indices of TCEs and EE, A23 declined steadily, while C21 and C22 fluctuated without significant increases over the last decade. Overall, tourism development in Lhasa still requires more green and low-carbon policies. All ED indices, except B31 and B32, presented fluctuating increases and then inflection points in 2019, after which growth weakened. In 2019, B31 and B32 decreased.

### 4.4. Verification of the Relationship between TCEs, EE, ED and Tourism Sustainability

The introduction and mechanism sections mention TCEs, EE and ED as three key aspects of sustainable tourism development. To validate their relevance to sustainable tourism development, the indices above were utilised and the degree of influence between these indicators and coordination was calculated. The results are shown in Table 6.

Following the principles that the larger the q statistic, the greater the impact of the factor on the results, and the smaller the *p*-value, the more significant the effect [67], indicators with q statistics greater than 0.9 and *p*-values less than 0.03 were selected. Nine indices met these criteria: A13, A21, A24, B11, B13, B16, B17, B22 and C11. The ED section had the most indices meeting the criteria, with several indices related to tourism revenue. Therefore, the analysis was summarised using B13, which is most directly related to tourism revenue. Additionally, traffic TCEs were the most significant contributor to total TCEs in the tourism industry, and the TCEs generated by traffic far exceeded those from accommodation and activities. A11 was used instead of A13 as the index for the TCEs category in the analysis. In the EE section, only one negative index met the criteria. To explore the contribution of environmental protection to tourism sustainability, the positive index C22 was included, which had a significant effect. Subsequently, a detailed analysis of the seven selected indices (A11, A21, A24, B11, B13, C11 and C22) was conducted.

**Table 6.** Degree of influence of each index on *D*.

| Type | q Statistic | *p*-Value | Type | q Statistic | *p*-Value |
|---|---|---|---|---|---|
| A11 (*Traffic TCEs*) | 0.76352 | 0.44060 | B15 (*Domestic tourist numbers*) | 0.50000 | 0.63157 |
| A12 (*Accommodation TCEs*) | 0.79096 | 0.01465 | B16 (*Tourism consumption per person*) | 0.93484 | 0.00294 |
| A13 (*Activity TCEs*) | 0.95432 | 0.00363 | B17 (*Total tertiary industry*) | 0.90176 | 0.00942 |
| A14 (*Total TCEs*) | 0.76352 | 0.44060 | B21 (*Total ED as a proportion of tertiary industry*) | 0.92994 | 0.03022 |
| A21 (*TCE intensity*) | 0.91714 | 0.01992 | B22 (*Total ED as a proportion of GDP*) | 0.92265 | 0.00831 |
| A22 (*TCE density*) | 0.76352 | 0.44059 | B31 (*Increase rate of ED*) | 0.30186 | 0.93135 |
| A23 (*Carbon production*) | 0.66492 | 0.16004 | B32 (*Growth rate of tourist number*) | 0.34729 | 0.92404 |
| A24 (*Per capita TCEs of tourists*) | 0.90402 | 0.01049 | C11 (*SO$_2$ emissions*) | 0.93443 | 0.00300 |
| B11 (*Tourist numbers in Lhasa*) | 0.91136 | 0.01647 | C12 (*Discharge of sewage*) | 0.48915 | 0.61578 |
| B12 (*Lhasa GDP*) | 0.88319 | 0.02796 | C21 (*Nature reserve number*) | 0.54667 | 0.85688 |
| B13 (*ED of Lhasa*) | 0.91863 | 0.02079 | C22 (*Percentage of forest cover*) | 0.67794 | 0.32251 |
| B14 (*Domestic tourism revenue*) | 0.45913 | 0.90063 | | | |

Except for A11 and C22, the *p*-values of other factors were less than 0.03, indicating that the test results for these factors were highly reliable. The q statistics for A21, A24, B11, B13 and C11 were greater than 0.9, indicating that these factors had a considerable impact on D. As an important part of TCEs in the Lhasa tourism industry, A11 also demonstrated a significant impact of 0.76. However, the influence of C22 was 0.68, indicating that the substantial cost of improving EE through afforestation would yield relatively modest results.

By comparing the five factors with q statistics greater than 0.9, it was found that C11 > B13 > A21 > B11 > A24. As indicators of environmental pollution, C11 ranked first and B11 ranked fourth. This confirmed the fragility of the EE of the plateau and that damage to the environment caused by human activities would greatly affect the coordination of the ecosystem. The second factor was B13. The development of tourism in Lhasa has generated substantial tourism income for the local area. This revenue serves as an economic foundation for the government to further promote green tourism, environmental management and technological innovation. A21 and A24 ranked third and fifth, respectively. A21 reflects the dependence of economic growth on TCEs to a certain extent, and A24 directly reflects the TCEs generated by tourists during their travel. The increase in TCEs will further aggravate the pressure on the EE. The high impact of A21 and A24 on the coordination degree highlights the importance of accelerating industrial upgrades and improving carbon utilisation. Additionally, it is particularly important to promote green tourism and reduce TCEs generated during travel.

The above test showed that the three aspects of TCEs, EE and ED were closely related to the sustainable tourism development.

*4.5. Coupling Relationship between Subsystems in Different Stages*

The coupling coordination model was applied to analyse the interactions among TCEs, ED and EE in Lhasa. As shown in Figure 6, from 2010 to 2020, there was an overall increasing trend in *D* among the three subsystems. The specific situation is described below.

The *C* of TCEs, ED and EE in Lhasa increased from 0.79 to 0.99 during 2010–2020. This transitioned from the 'running-in stage' to the 'high-level coupling stage', as per the classification in Table 3. *C* generally increased, albeit insignificantly at some points. The mean *C* was 0.92, with an average annual growth rate of 2.22%. In 2014, *C* decreased by 0.05. In 2012–2014, there was a notable surge in the growth rate, with the average annual increase reaching 9.8%. After 2015, the growth rate slowed significantly until 2017, when it surged to 3.94%. In 2016, there was a slight decline of 0.02.

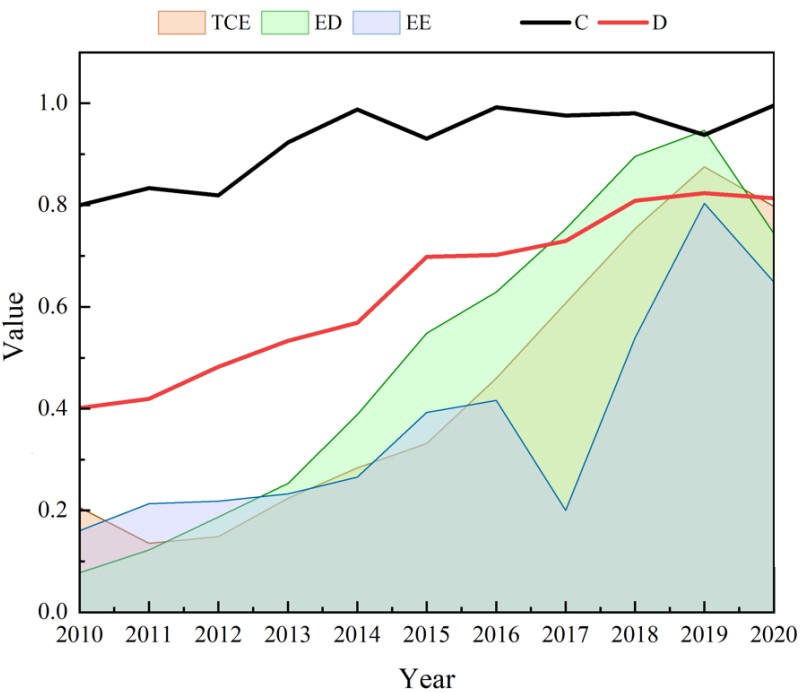

**Figure 6.** *C* and *D* values for TCEs, ED and EE (2010–2020).

*D* ranged from 0.40 to 0.82. Except for a slight decline in 2020, there was an overall increasing trend in *D*. The growth rate exhibited a consistent and stable trend from 2010 to 2014, fluctuating between 0.30 in 2010 and 0.56 in 2014. The average annual growth rate over this period was 9.1%. The peak growth rate occurred in 2015, with an increase of 0.13, followed by a decrease from 2016 to 2019. The average annual growth rate during this period was 5.5%. In 2020, the growth rate of *D* decreased by 0.01. In general, the *D* of TCEs, ED and EE from 2010 to 2012 ranged between 0.4 and 0.5, which was not a significant change. In the 'primary harmony stage' (as per the classification in Table 4), *D* exceeded 0.5 for the first time in 2013. Subsequently, between 2013 and 2015, *D* remained between 0.5 and 0.7. In 2016, *D* exceeded 0.7, entering the 'intermediate harmony stage'. Between 2018 and 2020, *D* remained robust, ranging from 0.8 to 0.9, entering the 'harmony stage' and meeting the requirements for sustainable development.

*4.6. Decoupling Relationship between ED and TCEs*

The decoupling relationship (Figure 7) between ED and TCEs from 2010–2020 was calculated by the Tapio model. The growth rates in ED and TCEs were positive, except for 2019 to 2020. Specifically, the growth rate in ED fluctuated upward during 2010–2014, with the growth rate increasing from 0.096 to 0.152. A turning point was observed in 2014–2015, and the growth rate plummeted by 0.17. Subsequently, the growth rate changed slightly until 2019 and dropped by 0.176 during 2019–2020. It then decreased for the first time (−0.072) in the past decade. The rate of variation in TCEs decreased gradually from 0.41 in 2010 to 0.152 in 2014 but increased from 0.152 in 2014 to 0.321 in 2016. However, it plummeted from 0.321 in 2016 to −0.54 in 2020, resulting in negative growth. The tourism economy has shown stable development since 2010, with revenue increasing each year. With the improvement in tourism facilities, the annual rate of increase in TCEs decreased. This suggests that the green tourism development measures advocated by Lhasa have yielded results [72].

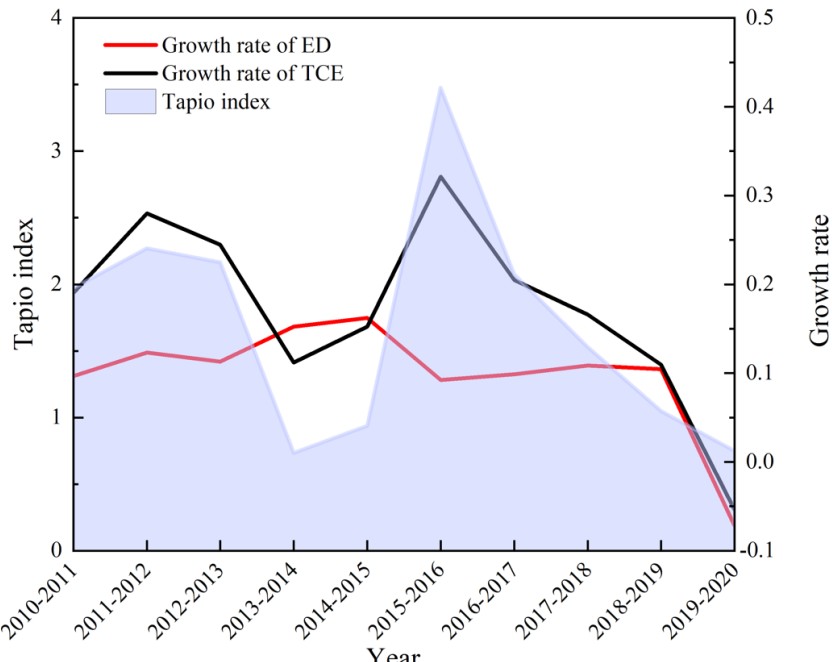

**Figure 7.** Variations in the Tapio index and change rates of ED and TCEs in Lhasa, 2010–2020.

The Tapio index generally shows a fluctuating downward trend. It decreased first, with a downward trend in overall volatility (2010–2014), then rose sharply (2014–2016), then declined (2016–2020). The carbon Tapio index decreased from 1.97 in 2010 to 0.74 in 2013. The decoupling relationship shifted from 'expansive negative decoupling' to 'weak decoupling' based on growth, as per the classification in Table 5. During the sharply increasing stage, the Tapio index rose by 2.74, shifting the decoupling relationship from 'weak decoupling' to 'expansive negative decoupling'. In the slow reduction stage, the Tapio index decreased by 2.73 from 3.48 in 2016 to 0.75 in 2020, and the decoupling relationship transitioned from 'expansive negative decoupling' to 'weak decoupling', based on growth. Overall, the Tapio index decreased annually except in 2014–2015.

The trends in tourism economic growth in Lhasa suggested a generally low dependence on TCEs. During 2014–2015, the Tourism Bureau of Lhasa issued several policy measures regarding tourism economic operations, public services and rural tourism, and continuously increased tourism infrastructure. These measures have driven tourism development in Lhasa and increased the total TCEs. The increase rate in the tourism economy peaked in 2015, which increased the dependence of ED on TCEs. In 2016, following intervention by the government of China, 'green' tourism and ecological protection measures were prioritised in tourism development [73]. This not only ensured a stable rate of increase in ED and a significant decrease in the growth rate of TCEs, but also decreased the dependence of ED on TCEs. In 2019–2020, tourist numbers decreased sharply, causing a dramatic plunge in tourism revenue and eventually decreasing the Tapio index. Overall, the Tapio index showed a downward trend, which indicated that Lhasa had reduced its dependence on TCEs while generating tourism revenue. It also shows that the region has taken a positive step towards the sustainable development of tourism.

## 5. Discussion

### 5.1. Main Findings

This study employed various EE, ED and TCE indices in conjunction with calculations of TCEs, the coupling coordination model and the Tapio decoupling model. The aim was to investigate the prospects for sustainable tourism development in Lhasa from a more comprehensive viewpoint. The results provide a new direction for exploring the sustainable development of plateau tourism. This work also explored changes in TCEs

related to different transportation modes. This provides decision-making support for a transition to low-carbon practices in plateau tourism.

It was found that sustainable tourism development was not only closely related to human tourism activities but was also affected by the environmental awareness of stakeholders, the national and local strategic layout, and policy guidance. This was because each inflection year of variations in the indicators of subsystems, *C* and the Tapio index corresponded to major national or local strategies and measures, or development measures of stakeholders. In 2012, the opening of the coastal route to Lhasa connected other cities with Lhasa, stimulating the tourism economy [70]. In 2014, the State Council vigorously promoted the improvement of tourism transportation services and supported the characteristic tourism industry through the proposal of tourism reforms and development [74]. As a result, the *C* of the three subsystems of TCEs, EE and ED decreased significantly and the Tapio index increased significantly during this period. In 2016, the Chinese government issued the 13th Five-Year Plan for the Development of the Tourism Industry. As a blueprint for China's social and economic development, the plan prioritised environmental protection and development and placed more emphasis on sustainable development [72]. Subsequently, the Tibetan government issued several tourism policies that attached importance to environmental protection [75]. As a result, the interaction among the three subsystems became more balanced and the development of tourism was less dependent on TCEs. Due to the COVID-19 pandemic in 2020, the tourist flow decreased, the tourism economy was adversely affected and *D* dropped sharply. Additionally, variations in the positive indices of TCEs and EE revealed that China's policies have had an important impact on the development of the tourism industry. Follow-up, intense and continuous implementation of environmental policies is crucial to sustainable tourism development.

The average *D* value of 0.63 indicated that the system was classified as 'primary coordinated'. This was because although China has developed rapidly in recent years, it is still a developing country. Rapid ED relies heavily on the transportation industry, which mainly uses fossil fuels [76]. This results in significant negative environmental impacts from economic activities. Thus, the development of the three subsystems has differed significantly. Meanwhile, shortcomings like poor overall coordination and endogenous power levels have led to poor coupling coordination [28]. The overall Tapio index showed a downward trend and reached 'weak decoupling' in 2020. This showed that although the dependence of ED on TCEs has decreased, it has not achieved complete independence. Therefore, urban planners should prioritise low-carbon sustainable development to implement a strong and sustainable development strategy. This would involve promoting energy upgrades and transformation, transitioning from fossil fuels to clean energy, and achieving long-term sustainable tourism development.

### 5.2. Limitations and Future Work

This work concentrated on examining the CCRs among TCEs, ED and EE in Lhasa, Tibet, in order to address the lack of research on plateau tourism and provide a basis for exploring its sustainable development. The study was subject to the following limitations. (1) Since the statistics of the Tibet Autonomous Region for the past three years have not been made public, data from 2010 to 2020 were used. There were some limitations on the time scale; future research could expand the time scale to obtain more up-to-date analyses. (2) Lhasa was used as a representative city of plateau tourism. Future studies could further explore the factors that affect the sustainable development of plateau tourism and deepen this into the investigation of other interesting mechanisms. (3) This paper focused on three subsystems of a coupling coordination model: TCEs, ED and EE. In the study of other issues, different subsystems could be analysed according to different needs. (4) The coupling coordination degree model and the Tapio model used in this paper may oversimplify the mechanisms of TCEs, ED and EE, and neglect social aspects. Additionally, a linear hypothesis was used to analyse their relationships, ignoring the influence of nonlinear

trends. In future studies, the influence of nonlinear trends and other external factors could be considered to more accurately represent the relationships among TCEs, EE and ED.

## 6. Conclusions

This study analysed TCEs stemming from tourism traffic, tourism accommodation and tourism activity in Lhasa during 2010–2020. The CCRs between TCEs, ED and EE were studied to determine the evolution characteristics of the coupling and the development relationship between tourism and EE. The decoupling relationship between TCEs and ED was also explored. Through this comprehensive analysis, some major conclusions were drawn.

(1) Variations in TCEs within the Tibetan Plateau region were closely related to the scale and intensity of tourism traffic, accommodation and activities. As the local tourism industry and the intensity of consumer activities increased, TCEs increased significantly and continuously. Among the three major types of carbon-emitting tourism activities, traffic was the most important. Additionally, as aircraft emit far more carbon per kilometre than trains and road vehicles, the proportion of total TCEs from aircraft was also significantly higher than that from trains and road vehicles. Therefore, it is essential to leverage government functions fully. This would entail opening additional railway and highway routes, enhancing public awareness of environmental conservation and opting for eco-friendly tourism. Simultaneously, it is imperative to expedite technological innovation, enhance carbon productivity and progressively realise sustainable development.

(2) TCEs, ED and EE are pivotal factors influencing the sustainable development of tourism. The *D* of these three factors peaked after ten years of development. The coupling coordination shifted from a 'reluctant' level in 2010 to a 'good' state in 2020, which met the conditions for sustainable tourism development. However, the average *D* of the three subsystems was only 0.63, which represents 'primary' coordination. Although the Tapio index showed a downward trend, the dependence of ED growth on TCEs decreased. The ED indices were significantly higher than the TCEs and EE indices. This showed that there is still room for improvement in the coordination between economic construction and environmental protection. The synergy between the three factors should be maximised to promote the sustainable development of tourism.

(3) National and local policies on ED are increasing the sustainability of tourism in the Tibetan Plateau. Due to its fragile environment, sustainable tourism development must prioritise ecological protection without compromise. Dynamic environmental assessments tailored to the geographic context of tourism industry development are crucial for formulating local tourism development plans. In the follow-up implementation process, legal supervision and restraint must be applied to ensure the sustainable development of plateau tourism through low-carbon tourism.

## 7. Main Policy Recommendations

Through calculating the TCEs in Lhasa and analysing the coupling relationship between ED and EE, the following policy recommendations are proposed to promote sustainable tourism development in the Tibetan Plateau.

(1) In the planning and development of tourism activities, government departments should promote green tourism and enhance environmental protection awareness. Tourists should also be strongly encouraged to choose trains or road vehicles to reduce TCEs. Although such measures require a lot more effort and money from the government, they are worth implementing.

(2) Local governments should introduce green tourism development enterprises and accelerate industrial development and technological reform to promote the development of clean energy and reduce the use of fossil fuels. The carbon utilisation rate of transportation, especially aircraft, should be improved. However, technological

change will not happen overnight, so new directions for energy optimisation also need to continue to be explored.

(3) Further advancement of the plateau ecosystem tourism industry should maintain a primary focus on ecological protection. Local tourism planners are urged to undertake dynamic environmental assessments. Fully considering the evolution and geographical background of tourism is necessary to achieve sustainable low-carbon tourism.

(4) New laws for ecological protection and supervision of the TCEs of enterprises and departments are needed. Reduction in TCEs can reduce damage to the environment and provide tourists with better scenery and experiences, as well as decrease the cost of environmental protection. The saved funds can be allocated to provide material subsidies to tourists who opt for low-carbon tourism. Additionally, they can offer economic support to local governments and enterprises that effectively implement low-carbon activities, encouraging them to participate in the construction of sustainable tourism development.

**Author Contributions:** Conceptualisation, L.Y. and L.C.; Methodology, L.Y., J.W., L.C., Y.H. and X.Y.; Writing—original draft preparation, L.Y., J.W. and X.Y.; Writing—review and editing, L.Y., J.W., L.C., Q.Z. and Y.H.; Visualisation, J.W., Y.H. and X.Y. All authors have read and agreed to the published version of the manuscript.

**Funding:** This research was funded by the Humanities and Social Sciences Research Project of the Ministry of Education, grant number 18YJCZH012 (funder: the Ministry of Education), and the Young and Middle-Aged Academic Leaders of "Blue Project" in Jiangsu Universities (2022–2025), grant number SuJiaoShiHan [2022] 29 (funder: Jiangsu Provincial Department of Education) and the Academic Top Talent Training Project of the Jinling Institute of Technology (2020–2023), grant number JinYuanZi [2020] 8 (funder: Jinling Institute of Technology).

**Institutional Review Board Statement:** Not applicable.

**Informed Consent Statement:** Not applicable.

**Data Availability Statement:** The data used to support the findings of this study are available from the corresponding author upon request.

**Acknowledgments:** The author expresses sincere gratitude to the editors and anonymous reviewers for their valuable suggestions.

**Conflicts of Interest:** The authors declare that they have no known competing financial interests or personal relationships that could have appeared to influence the work reported in this paper.

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
