# Peer review of "Coupling and Coordination between Tourism, the Environment and Carbon Emissions in the Tibetan Plateau"

_sustainability, doi:10.3390/su16093657_

Round 1
Reviewer 1 Report
Comments and Suggestions for Authors
0. Regarding the originality of the research, this manuscript fell short. A significantly high 38% similarity for respectable original research was identified by the Turnitin software. The bare minimum margin typically demanded by a publisher is 20-25%. In contrast, particularly within the realm of sustainability, the sustainability journal is the preeminent publication. Insufficient originality characterized the manuscript.
1. Please see the following suggestions to improve the manuscript.
|
Section |
Observations & Suggestions for Improvement |
|
Title |
The current title is informative but could be more succinct to captivate the reader's interest effectively. An enhancement proposal is to guarantee that the title concisely captures the primary emphasis of the study. For example, the manuscript "Tourism, Ecology, and Carbon Emissions in Tibetan Plateau: A Decoupling and Coordination Analysis" provides a more accurate and persuasive summary of its content. |
|
Abstract |
The concise and enlightening abstract provides a clear overview of the study's extent, approach, and significant discoveries. Nevertheless, the manuscript would be improved by providing a more precise and direct statement of the study's primary objective or research question to communicate the focus of the research to the reader immediately. |
|
Introduction |
The introduction offers a comprehensive summary of the importance of tourism in the Tibetan Plateau, the environmental obstacles it faces, and the research framework in which it is situated. To enhance its quality, the study must provide a more explicit and precise definition of the problem or deficiency it aims to tackle in the current research. Adding a more comprehensive elucidation of the study's aims and a concise methodology summary would further improve this section. |
|
Methodology |
The methodology section, particularly the study area description, is effectively expressed. Nevertheless, the manuscript would be enhanced by a more comprehensive elucidation of the chosen methodologies, specifically regarding their appropriateness for analyzing the interconnection and synchronization between the variables under investigation. This section would benefit from a clear rationale for selecting models and their applicability to the specific context, as it would enhance its overall strength. |
|
Results |
The results section adeptly presents the findings, demonstrating distinct patterns and points of change. Nevertheless, it is possible to establish a more immediate correlation between the outcomes and the research aims. To enhance the effectiveness of this section, it is crucial to ensure that the results directly correspond to the research questions and objectives. |
|
Discussion |
The discussion offers a comprehensive analysis of the study's findings within the framework of current literature and the potential impact on policies. To enhance its quality, the focus should be directed towards elucidating how these findings contribute to novel insights or viewpoints within sustainable tourism. Furthermore, establishing a direct correlation between the points of discussion and the study's objectives would improve the overall coherence and relevance. |
|
Conclusion |
The conclusion aptly encapsulates the study's findings and their consequential significance. To enhance its quality, the text should concisely summarize the findings' theoretical, practical, and policy ramifications. Additionally, considering the study's limitations and findings, offering recommendations for future research would contribute to a more thorough conclusion. |
|
English Usage |
The manuscript could be better crafted. However, improving grammatical accuracy, particularly in using articles, punctuation, and commas and simplifying complex sentences, would enhance clarity and readability. Enhancing sentence conciseness and ensuring a clear presentation of ideas will improve the manuscript's accessibility to many readers. To enhance the overall quality of the manuscript, it is crucial to ensure that there are no run-on sentences and to review for common grammatical errors thoroughly. For instance, the abstract contained multiple phrases: "There were ... model, ... model, ... model....". This sentence lacks professionalism. |
|
Questionable reliability and validity of this study |
The manuscript did not adhere to the formatting guidelines of MDPI. The manuscript mentioned above solely comprised a compilation of statistical data. No notable advancements or contributions in sustainability, travel, statistics, machine learning, and environments existed. Several citations in the introduction sections needed to be updated, exceeding five years. Why is it necessary to estimate historical data rather than future data? In section 3.3, numerous recent studies have shown that accurate and improved predictions can be made regarding the future benefits of procedures, plans, and policies. |

The manuscript could be better crafted. However, improving grammatical accuracy, particularly in using articles, punctuation, and commas and simplifying complex sentences, would enhance clarity and readability. Enhancing sentence conciseness and ensuring a clear presentation of ideas will improve the manuscript's accessibility to many readers. To enhance the overall quality of the manuscript, it is crucial to ensure that there are no run-on sentences and to review for common grammatical errors thoroughly. For instance, the abstract contained multiple phrases: "There were ... model, ... model, ... model....". This sentence lacks professionalism.
Author Response
Dear editor and reviewers ,
Thank you for the reviewers ' careful review and valuable comments. We sincerely appreciate your suggestion and have carefully considered every opinion. In response to the comments made, we have revised the title, abstract, introduction, method, results, discussion and conclusion of the article, and added the limitations of the article and policy recommendations. In addition, for the continuity of the article, the analysis of changes in total carbon emissions is added in the results section. The specific responses are as follows:
Reviewer 1:
(1) The current title is informative but could be more succinct to captivate the reader's interest effectively. An enhancement proposal is to guarantee that the title concisely captures the primary emphasis of the study. For example, the manuscript "Tourism, Ecology, and Carbon Emissions in Tibetan Plateau: A Decoupling and Coordination Analysis" provides a more accurate and persuasive summary of its content.
Response: Thank you for your suggestions. According to your advice, the title has been changed.
(2) The concise and enlightening abstract provides a clear overview of the study’s extent, approach, and significant discoveries. Nevertheless, the manuscript would be improved by providing a more precise and direct statement of the study’s primary objective or research question to communicate the focus of the research to the reader immediately.
Response: Thank you for your suggestions. We have rewritten the abstract according to your suggestion, further clarified research question and objective of the paper.
(3) The introduction offers a comprehensive summary of the importance of tourism in the Tibetan Plateau, the environmental obstacles it faces, and the research framework in which it is situated. To enhance its quality, the study must provide a more explicit and precise definition of the problem or deficiency it aims to tackle in the current research. Adding a more comprehensive elucidation of the study's aims and a concise methodology summary would further improve this section.
Response: Thank you for your suggestions. The introduction has been reorganized according to the recommendations.
(4) The methodology section, particularly the study area description, is effectively expressed. Nevertheless, the manuscript would be enhanced by a more comprehensive elucidation of the chosen methodologies, specifically regarding their appropriateness for analyzing the interconnection and synchronization between the variables under investigation. This section would benefit from a clear rationale for selecting models and their applicability to the specific context, as it would enhance its overall strength.
Response: Thank you for your suggestions. According to your recommendation, the description of methods and the links between methods has been added to each method.
(5) The results section adeptly presents the findings, demonstrating distinct patterns and points of change. Nevertheless, it is possible to establish a more immediate correlation between the outcomes and the research aims. To enhance the effectiveness of this section, it is crucial to ensure that the results directly correspond to the research questions and objectives.
Response: Thank you for your suggestions. According to your opinions, the result part was modified, and the description of the total amount of carbon emissions was added to increase the connection with the research purpose.
(6) The discussion offers a comprehensive analysis of the study's findings within the framework of current literature and the potential impact on policies. To enhance its quality, the focus should be directed towards elucidating how these findings contribute to novel insights or viewpoints within sustainable tourism. Furthermore, establishing a direct correlation between the points of discussion and the study's objectives would improve the overall coherence and relevance.
Response: Thank you for your suggestions. The discussion section has been revised according to the opinions, which enhances the coherence between the main points of the discussion and the objectives of the study.
(7) The conclusion aptly encapsulates the study's findings and their consequential significance. To enhance its quality, the text should concisely summarize the findings' theoretical, practical, and policy ramifications. Additionally, considering the study's limitations and findings, offering recommendations for future research would contribute to a more thorough conclusion.
Response: Thank you for your suggestions. The conclusion section has been simplified according to the recommendations, the policy recommendations are placed in the next chapter separately, and the limitations of this article and the recommendations for future research are placed in the discussion section.
(8) The manuscript could be better crafted. However, improving grammatical accuracy, particularly in using articles, punctuation, and commas and simplifying complex sentences, would enhance clarity and readability. Enhancing sentence conciseness and ensuring a clear presentation of ideas will improve the manuscript's accessibility to many readers. To enhance the overall quality of the manuscript, it is crucial to ensure that there are no run-on sentences and to review for common grammatical errors thoroughly. For instance, the abstract contained multiple phrases: "There were ... model, ... model, ... model....". This sentence lacks professionalism.
Response: Thank you for your suggestions. According to your advice, this manuscript was edited for proper English language, grammar, punctuation, spelling, and overall style by one or more of the highly qualified native English speaking editors at NativeEE. NativeEE specializes in editing and proofreading scientific manuscripts for submission to peer-reviewed journals.
- The manuscript did not adhere to the formatting guidelines of MDPI. The manuscript mentioned above solely comprised a compilation of statistical data. No notable advancements or contributions in sustainability, travel, statistics, machine learning, and environments existed. Several citations in the introduction sections needed to be updated, exceeding five years. Why is it necessary to estimate historical data rather than future data? In section 3.3, numerous recent studies have shown that accurate and improved predictions can be made regarding the future benefits of procedures, plans, and policies.
Response: Thank you for your suggestions. The references of the introduction have been updated according to the recommendations. Because the development of tourism in the Qinghai-Tibet Plateau is not long, this paper aims to use past data to explore the relationship between tourism and ecological environment, in order to analyze whether the tourism industry in the Qinghai-Tibet Plateau is sustainable, and provide reference and suggestions for future tourism planning and development.

Reviewer 2 Report
Comments and Suggestions for Authors
This article studies coupling relationship in Lasha, showing that government internvention does improve coordination between tourism and nature, but more is said than done, thus calling for more sustainable planning in future.

See enclosed.
Author Response
Dear editor and reviewers ,
Thank you for the reviewers ' careful review and valuable comments. We sincerely appreciate your suggestion and have carefully considered every opinion. In response to the comments made, we have revised the title, abstract, introduction, method, results, discussion and conclusion of the article, and added the limitations of the article and policy recommendations. In addition, for the continuity of the article, the analysis of changes in total carbon emissions is added in the results section. The specific responses are as follows:
Reviewer 2:
(1)” both Chinese and foreign tourists” for "all" tourists since Chinese should not be emphasized in an international journal.
Response: Thank you for your suggestions. It has been changed to ' visitor'.
(2) “home and abroad. Although government departments have constructed an ecological civilization from the perspective of environmental protection, identifying the bottom line, implementation of ecological projects, mineral resource development and high-pollution enterprise limitations (Sun, Zheng, Yao, & Zhang, 2012). Nevertheless, less attention has been paid to how tourism impacts the environment (Yan, Zhang, & Shan, 2014).” This reflexive clause should be connected in one sentence, rather than two separated by a full-stop.
Response: Thank you for your suggestions. The sentence has been reorganized as suggested.
(3) “steadily” Do you mean "rapidly"?
Response: Thank you for your suggestions. Yes, the text has been modified.
(5) “However, these studies are mainly focused on areas with advanced tourism development, such as Chengdu-Chongqing, Yangtze River Economic Belt, Northeast
China, and coastal areas in China.” Please do not limit your literature review to China, and please cite a few from other countries as well.
(6)” material” Please specify what kind of materials here
Response: Thank you for your suggestions. What is expressed here is that economic development provides economic support for promoting industrial upgrading of carbon emissions and measures needed to protect the environment.
Response: Thank you for your suggestions. The statement has been changed.
(8) “by, economic development via” This is convoluted; do you mean economic development serves as a mediating variable?
Response: Thank you for your suggestions. Now the article has been modified, the revised version does not have this sentence.
Response: Thank you for your suggestions. Now the article has been modified, the revised version does not have this sentence.
(10) “decreased expenditure” How does decreased expenditure connote better environmental performance? This is more speculative than deduced, which is not rigorous at all.
(11)” low cost” It may be higher cost instead, and this is by and large uncorroborated.
Response: Thank you for your suggestions. What we want to express here is that the less money spent on travel, the more people will be attracted to travel, and the increase in the number of tourists will promote the increase of the tourism economy.
(12)” 29655.5 km2” You must use thousand-digit separator
Response:Thank you for your suggestions. Thousand separators have been used as recommended.
(13)”3650m” ditto
Response: Thank you for your suggestions. Thousand separators have been used as recommended.
(16)” the official website of the Lhasa Tourism Bureau (https://lfj.lasa.gov.cn/lsslyfzj/zwgk/zwgk.shtml) during 2010-2020.” The website should be cited in the appendix rather than in-text.
(18) ”365” Technically, should it be 365.25?
Response: Thank you for your suggestions. I consulted other people 's literature and they also use 365.
(19)” 155 MJ/bed based on existing studies (Shi & Wu, 2011; Gössling, 2002)” Can you do a meta-analysis of those numbers before you employ them
Response: Thank you for your suggestions. We added the reason use these numbers. This is based on the energy consumption coefficient per bed of star-rated hotels calculated by energy consumption and hotel occupancy rate, which is based on the previous research results.
Response: Thank you for your suggestions. We have revised. The coefficient has been one-to-one correspondence with the activity type.
(21)” Since TCE, ED and EE are equally important, α=β=γ=1/3..” Please endogenize those weights
Response: Thank you for your suggestions. By consulting the literature, it is found that each subsystem is regarded as equally important when calculating T. In this paper, there are three subsystems, so the value obtained here is 1 / 3.
(22) “U1, U2 and U3 represent the coordination evaluation values of the three subsystems of TCE, ED, and EE” Move this immediately to the sentence before to show clear connections.
Response: Thank you for your suggestions. We have revised them. The abbreviations have been corresponded to the corresponding subsystems one by one.
(23) “(8)” From the descriptions below, do you mean m = 8, n = 11, and k = 4?
Response: Thank you for your suggestions. Yes, but k = 3, because the EE subsystem has only three factors, It has been explained in the text.
Response: Thank you for your suggestions. References have been added.
(26) Can you use horizontal dotted grid lines in order to assort the items into subcategories more clearly?
Response: Thank you for your suggestions. We have modified it. The second and third levels of the table have been separated by lines.
(28) “2” make sure that subscripts appear appropriately everywhere
Response: Thank you for your suggestions. All the places that need to be subscripted in the full text have been verified and modified.
Response: Thank you for your suggestions. The English names of the file are indeed capital letters.
Response: Thank you for your suggestions. In Chinese, endogenous power refers to the spontaneous power generated by the needs of survival and development within the organization. In this paper, it is said that there is no inherent development power within the subsystem, which affects the overall coupling coordination degree.
Response: Thank you for your suggestions. Managers here refer to urban planners and managers.

Reviewer 3 Report
Comments and Suggestions for Authors
Dear authors,
Your research is interesting, but still it must do something to ensure a good quality of the research:
- In the introduction part you must mention the hypothesis of the research.
- In the methodology part you must add the limitation of the study and the actions for remove this limitations on the hypothesis proposed
- Before conclusion part it would be appropriate to introduce a discussion section, here you could come with your own proposals, based on your study regarding the future direction of sustainable development of tourism in researched area.
- As far as concerning the reference part you may consider appropriate to add other references, as it is now are not enough!!
Reviewer 4 Report
Comments and Suggestions for Authors
The topic selected for this paper is interesting and it is capable of exploring some trends in the factors that represent some of the most relevant elements in the discussion about sustainability. The authors identified a field and an ecosystem that deserves to be paid attention to there is great potential to see how sustainability is contributing to the development of the area economically but also how this development can be controlled if we limit our impact. This is very good, and I strongly encourage the authors to pursue further research in the field. However, I see space for improvements that could bring the paper to a better level, overcoming the limits I see related to the academic contribution and the implications for the industry. These are some suggestions I would like to give to the authors.
- First of all, in the abstract, the authors clarify what is their goal, but it is unclear what is the novelty of the findings and the actual contribution to academia and industry. This can be more imagined than read through the entire document. Why this is a relevant study that could contribute to academia and practitioners’ knowledge? Please make it clearer.
- In particular, it is not clear how and why it is relevant to discuss the questions identified by the authors.
- There is somehow a justification for the fact that limited studies have been done on the plateau but why it is important to care about the studies on the plateau is important? There is a vague justification that relates to the fact that there is a fragile ecosystem. This is correct, but the authors should develop more this argument.
- Why did you consider the three factors you have discussed and not others? Sustainability is all about emissions. Sustainability is only about the environment and economic development. Of course not, so you should justify your choices more.
- In addition to this, I would suggest authors to dig further into the literature related to fragile environments. If limited literature might be found for the plateau more relevant literature could be used to contextualise the work taking into consideration the perspective of other fragile environments. The literature part is very limited and weak.
- While reading the paper it is clear that it has potential but the literature review part is limited and it gives for granted several aspects of the discussion.
- I would suggest starting this discussion with a broader perspective that takes into consideration the discussion related to the general sustainability debate around the world, and then narrowing down the areas of interest. An academic paper is a document read by experts but it could be also read by people who are approaching the topic so we cannot take context for granted. Start from a broader perspective setting briefly the scenario, then move into the more specific aspects related to TCE, ED and EE and give more details on it before moving on to your main focus. Then discuss the coupling and decoupling aspects providing more information about what they are and why they are relevant
- Second, the choice of the focus of the research must be justified further. It looks like sustainability only has to do with the factors identified by the authors. Sustainability is not only about it, it is not only about the environment and the profits but also the people. And within the environment, it is not only about carbon emission. So why is the focus on these elements? It is correct but the reader while reading the document doesn’t get it. Explain.
- I would recommend pointing out the actual research questions and why they are relevant
- More specifically p. 1:
- Add some references to this statement “Further, the development of tourism in the Tibetan Plateau is still in its infancy, currently”
- Rewrite this statement, it is not clear: “Although government departments have constructed an ecological civilization from the perspective of environmental protection, identifying the bottom line, implementation of ecological projects, mineral resource development and high-pollution enterprise limitations (Sun, Zheng, Yao, & Zhang, 2012). Nevertheless, less attention has been paid to how tourism impacts the environment (Yan, Zhang, & Shan, 2014)”.
- “Unreasonable resource development and excessive tourism activities have increased the pressure on the ecological environment (Huang & Xin, 2022).”. What are the consequences of this? Why do we have to care about the fact that there has been pressure on the ecological environment? What is your perspective?
- “Nevertheless, less attention has been paid to how tourism impacts the environment (Yan, Zhang, & Shan, 2014)”: what about now - 10 years after? Do you have fresher references to add here?
- Where is this happening, on the plateau or not? Not clear:
- “Carbon dioxide emissions generated by human activity including vehicular traffic, accommodation and tourism have increased steadily, resulting in intense negative impacts including environmental pollution and ecological damage (Weng, Li, Yang, & Li, 2021).”
- More specifically p. 2:
- Rewrite this sentence it is not clear “The relationship between tourism carbon emissions and economic growth has been established (Weng, Li, Yang, & Li, 2021; Tang, Shang, & Shi 2014); Anthony (2017), coupling and coordination relationship of urban tourism-city-eco-environment system (Xie et al., 2021; Wang & Du, 2020; Pan, Weng & Li 2021).”
- “In this study, the CCR between TCE, ED, and EE in Lhasa has been analyzed to correlate plateau tourism with ecological environment.” Justify why we should focus on this, there is limited support for your analysis
- References for this statement is requested: “First, the environment is an essential factor for human production and life and is the basis for the development of tourism economy and related emissions.”
- Justify this with references related to it: “Secondly, the development of tourism economy is an inexhaustible motive force to slow down carbon emissions and ensure environmental protection.” For very long period of time the opposite happened.
- Add a reference: “Thus, an increase in the carbon utilization ratio in the tourism industry and decreased carbon emissions can lead to decreased expenditure. “
- “With the acceleration of economic development, many regions have shown a trend of sustainable tourism development. For example, in areas with good ecological environment and carbon emissions, green and low-carbon standards have been achieved; in areas with prosperous economic development and good ecological environment, it has met the requirements of harmonious environmental protection; in areas where economic development is prosperous and carbon emissions are up to standard, technological innovation and industrial upgrading have been achieved. However, in order to realize the sustainable development of tourism, we should not only ensure the coordinated development of tourism economy and low carbon emissions, but also take into account the protection of the environment. Only the coupling and coordinated development of these three can truly realize the sustainable development of tourism.“. This part does not have any references. On the basis of which source can you conclude this? Add literature evidence with proper references and justify more the dynamics that lead to these mechanisms
- Moving on to the research design and methodology there is some overview of what the authors have chosen to do, but I would recommend to justify more why they chose this methodology and not other methodology to address their analysis. What are the research questions and how you address them with your analysis should be better explained
- In particular:
- I would be very careful in showing where you took the data from. I would suggest developing a clear table where the authors show what the items under investigation and where they come from. The paragraph 3.2 is not clear.
- Justify why you took into consideration data from 2010 and 2020 and how you considered the data during the Covid period where of course your indicators have shown very low levels. This was due to a specific circumstance that none of us could have expected and this could generate a bias in the analysis
- In paragraph 3.3 the authors discuss their methodology to calculate the different variables relevant to the discussion. It would be good to explain more explicitly what authors are about to discuss and why those are relevant: after they can show the calculations
- In 3.3.2 do you assume that there is full occupancy every day of the year? Why?
- In 3.3.3 “According to previous studies, tourism activities were divided into sightseeing, leisure, business, family visits, and others. Based on the existing statistical path of the Tourism Sampling Survey Data, tourists were divided into urban and rural residents using the formula (Wei et al., 2012)” Explain why for your analysis it is important to divide between urban and rural residents
- In 3.3.3. you give the following “According to existing studies (Shi & Wu, 2011), carbon emission coefficients were 417, 1670, 786, 591 and 172 g/person.” Could you specify what these numbers refer to? Do they refer to the previous categories you mentioned?
- In 3.4.1 discuss more in depth the Tapio Decoupling model. Give also a definition of decoupling and coupling. The authors speak a lot about this but there is not a proper section where these concepts and their relevance is discussed. Authors might add this in the literature for example.
- How did you do this: “According to above characteristics of indices change, the tourism development of Lhasa in the last 10 years can be divided into three stages: smooth (2010-2015), rapid (20152019) and negative (2019-2020).” What were your assumptions to classify?
- In 5. Try to give a more in-depth explanation of your results. E.g. “ Although the coupling degree (average 0.93) between TCE, ED and EE in Lhasa is high, which is an important condition for the sustainable development of tourism, the average coupling coordination degree is only 0.63, which is at a primary coordinated level. There is still a lot of room for improvement.” Further elaboration on this is important. What do you want to say? Improvement in which direction?
- p. 13 Are the differences between the variation in carbon emission due to traffic, accommodation and activities statistically significant?
- In 5 & 6 you try sometimes to give explanations of some results without having evidence, neither from your analysis nor from the literature, please make sure to find some evidence for these statements. E.g. “ Results show that in 2014-2015, government policies accelerated the construction of tourism facilities, strengthened environmental protection, and rapidly increased environmental indicators, resulting in a decline in coupling. “. How do you know that it is the government policies that lead to these results? Did you test? Or this is your opinion? Do you have literature backing up your point of view? It would be very nice to have a statistically sound analysis that could help you not only to give a picture of the trends in all your indices but also to find out what could be the reasons for those changes. I would suggest the authors to create a cause-and-effect model to test some hypotheses related to the causes of the results obtained.
- In 5&6 you present more the what you found rather than showing the implications of all this analysis. These parts should be backed up with literature review evidence, and the implications should be supported by additional literature that maybe wasn’t covered in the literature review but that could help support your discussion. In addition to this, it would be very important to clearly show what is your contribution to academia and the industry. This contribution is not very visible. I would like to suggest to the author to further develop this point to be capable to show more the contribution coming from this work.
- In general, tables should be reformatted. For example, it is not clear to which second-grade indices the third-grade indices belong to: add proper lines to distinguish
- In Table 3 and 4 explain the phases and the sub-classes
- Tell in 4. where the reader can find the results: i.e. Figure 3
- In the work, there is no limitation of the research and no future of the research. I would like to suggest to the authors to include this part to reflect on the process and identify opportunities for improvement but also to give scholars and idea for further development
Author Response
Dear editor and reviewers ,
Thank you for the reviewers ' careful review and valuable comments. We sincerely appreciate your suggestion and have carefully considered every opinion. In response to the comments made, we have revised the title, abstract, introduction, method, results, discussion and conclusion of the article, and added the limitations of the article and policy recommendations. In addition, for the continuity of the article, the analysis of changes in total carbon emissions is added in the results section. The specific responses are as follows:
Reviewer 4:
Response: Thank you for your suggestions. The abstract part has been modified according to the recommendations.
(3) There is somehow a justification for the fact that limited studies have been done on the plateau but why it is important to care about the studies on the plateau is important? There is a vague justification that relates to the fact that there is a fragile ecosystem. This is correct, but the authors should develop more this argument.
(4) Why did you consider the three factors you have discussed and not others? Sustainability is all about emissions. Sustainability is only about the environment and economic development. Of course not, so you should justify your choices more.
Response: Thank you for your suggestions. TCE, ED and EE are inextricably linked, and are closely related to tourism. Therefore, it is assumed that these three are the key factors affecting the sustainable development of tourism, and they are verified in the subsequent text. The more detailed reasons for the selection are explained in the 2.2 part of the text.
(5) In addition to this, I would suggest authors to dig further into the literature related to fragile environments. If limited literature might be found for the plateau more relevant literature could be used to contextualise the work taking into consideration the perspective of other fragile environments. The literature part is very limited and weak.
(6) While reading the paper it is clear that it has potential but the literature review part is limited and it gives for granted several aspects of the discussion.
Response: Thank you for your suggestions. Some literature has been added according to the recommendations.
(7) I would suggest starting this discussion with a broader perspective that takes into consideration the discussion related to the general sustainability debate around the world, and then narrowing down the areas of interest. An academic paper is a document read by experts but it could be also read by people who are approaching the topic so we cannot take context for granted. Start from a broader perspective setting briefly the scenario, then move into the more specific aspects related to TCE, ED and EE and give more details on it before moving on to your main focus. Then discuss the coupling and decoupling aspects providing more information about what they are and why they are relevant
Response: Thank you for your suggestions. The introduction part has been revised according to the suggestions. Starting from the beautiful but fragile Qinghai-Tibet Plateau, the relationship between tourism and ecological environment is introduced, and finally the research methods are discussed.
(8) Second, the choice of the focus of the research must be justified further. It looks like sustainability only has to do with the factors identified by the authors. Sustainability is not only about it, it is not only about the environment and the profits but also the people. And within the environment, it is not only about carbon emission. So why is the focus on these elements? It is correct but the reader while reading the document doesn’t get it. Explain.
Response: Thank you for your suggestions. The beautiful scenery of the Qinghai-Tibet Plateau has attracted a large number of tourists, but because of its fragile ecological environment, it is considered that it is difficult to heal after the destruction of activities, and the Qinghai-Tibet Plateau plays an important role in the monitoring of global climate change, so it is of great significance to study the coordinated development of tourism development and environment. More detailed explanations have been made in this paper.
(9) I would recommend pointing out the actual research questions and why they are relevant
Response: Thank you for your suggestions. The reason why the computational decoupling has been coupled and coordinated has been explained in the method part.
(10) Add some references to this statement “Further, the development of tourism in the Tibetan Plateau is still in its infancy, currently”
Response: Thank you for your suggestions. Relevant literature has been added
(11) Rewrite this statement, it is not clear: “Although government departments have constructed an ecological civilization from the perspective of environmental protection, identifying the bottom line, implementation of ecological projects, mineral resource development and high-pollution enterprise limitations (Sun, Zheng, Yao, & Zhang, 2012). Nevertheless, less attention has been paid to how tourism impacts the environment (Yan, Zhang, & Shan, 2014)”.
(12) “Unreasonable resource development and excessive tourism activities have increased the pressure on the ecological environment (Huang & Xin, 2022).”. What are the consequences of this? Why do we have to care about the fact that there has been pressure on the ecological environment? What is your perspective?
Response: Thank you for your suggestions. The consequence of doing so is that the already fragile ecological environment of the Qinghai-Tibet Plateau will become more vulnerable, so we must pay enough attention and even take measures to intervene. There is a more detailed description in the article.
(13) “Nevertheless, less attention has been paid to how tourism impacts the environment (Yan, Zhang, & Shan, 2014)”: what about now - 10 years after? Do you have fresher references to add here?
Response: Thank you for your suggestions. Follow-up people still pay less attention to the impact of tourism on the environment, and new literature has been added.
(14) Where is this happening, on the plateau or not? Not clear: “Carbon dioxide emissions generated by human activity including vehicular traffic, accommodation and tourism have increased steadily, resulting in intense negative impacts including environmental pollution and ecological damage (Weng, Li, Yang, & Li, 2021).”
(15) Rewrite this sentence it is not clear “The relationship between tourism carbon emissions and economic growth has been established (Weng, Li, Yang, & Li, 2021; Tang, Shang, & Shi 2014); Anthony (2017), coupling and coordination relationship of urban tourism-city-eco-environment system (Xie et al., 2021; Wang & Du, 2020; Pan, Weng & Li 2021).”
Response: Thank you for your suggestions. This sentence has been restructured as suggested.
(16) “In this study, the CCR between TCE, ED, and EE in Lhasa has been analyzed to correlate plateau tourism with ecological environment.” Justify why we should focus on this, there is limited support for your analysis
(17) References for this statement is requested: “First, the environment is an essential factor for human production and life and is the basis for the development of tourism economy and related emissions.”
Response: Thank you for your suggestions. Relevant literature has been added.
(18) Justify this with references related to it: “Secondly, the development of tourism economy is an inexhaustible motive force to slow down carbon emissions and ensure environmental protection.” For very long period of time the opposite happened.
Response: Thank you for your suggestions. Relevant literature has been added.
(19) Add a reference: “Thus, an increase in the carbon utilization ratio in the tourism industry and decreased carbon emissions can lead to decreased expenditure. “
Response: Thank you for your suggestions. Reference has been added
(20) “With the acceleration of economic development, many regions have shown a trend of sustainable tourism development. For example, in areas with good ecological environment and carbon emissions, green and low-carbon standards have been achieved; in areas with prosperous economic development and good ecological environment, it has met the requirements of harmonious environmental protection; in areas where economic development is prosperous and carbon emissions are up to standard, technological innovation and industrial upgrading have been achieved. However, in order to realize the sustainable development of tourism, we should not only ensure the coordinated development of tourism economy and low carbon emissions, but also take into account the protection of the environment. Only the coupling and coordinated development of these three can truly realize the sustainable development of tourism.” This part does not have any references. On the basis of which source can you conclude this? Add literature evidence with proper references and justify more the dynamics that lead to these mechanisms
Response: Thank you for your suggestions. Relevant literature has been added in the paper.
(21) Moving on to the research design and methodology there is some overview of what the authors have chosen to do, but I would recommend to justify more why they chose this methodology and not other methodology to address their analysis. What are the research questions and how you address them with your analysis should be better explained
(22) I would be very careful in showing where you took the data from. I would suggest developing a clear table where the authors show what the items under investigation and where they come from. The paragraph 3.2 is not clear.
(23) Justify why you took into consideration data from 2010 and 2020 and how you considered the data during the Covid period where of course your indicators have shown very low levels. This was due to a specific circumstance that none of us could have expected and this could generate a bias in the analysis
Response: Thank you for your suggestions. In 2010-2020, the development of Lhasa 's tourism industry was very rapid, and there was a clear inflection point. Considering that the number of people affected by the epidemic plummeted 20 years later, it was only intercepted until 2020. In 2019, the growth rate of tourism industry changed from positive to negative for the first time. During this period, the main consideration was the decrease in the number of travelers caused by the new Covid epidemic, so the growth rate of tourism industry slowed down.
(24) In paragraph 3.3 the authors discuss their methodology to calculate the different variables relevant to the discussion. It would be good to explain more explicitly what authors are about to discuss and why those are relevant: after they can show the calculations
Response: Thank you for your suggestions. According to the recommendations, the description of three methods of carbon emissions, decoupling index and coupling coordination relationship and the relationship between the three relationships have been increased.
(25) In 3.3.2 do you assume that there is full occupancy every day of the year? Why?
(26) In 3.3.3 “According to previous studies, tourism activities were divided into sightseeing, leisure, business, family visits, and others. Based on the existing statistical path of the Tourism Sampling Survey Data, tourists were divided into urban and rural residents using the formula (Wei et al., 2012)” Explain why for your analysis it is important to divide between urban and rural residents
(27) In 3.3.3. you give the following “According to existing studies (Shi & Wu, 2011), carbon emission coefficients were 417, 1670, 786, 591 and 172 g/person.” Could you specify what these numbers refer to? Do they refer to the previous categories you mentioned?
Response: The coefficient has been one-to-one correspondence with the activity type.
(28) In 3.4.1 discuss more in depth the Tapio Decoupling model. Give also a definition of decoupling and coupling. The authors speak a lot about this but there is not a proper section where these concepts and their relevance is discussed. Authors might add this in the literature for example.
Response: Thank you for your suggestions. The definition of decoupling has been described in the method section.
(29) How did you do this: “According to above characteristics of indices change, the tourism development of Lhasa in the last 10 years can be divided into three stages: smooth (2010-2015), rapid (20152019) and negative (2019-2020).” What were your assumptions to classify?
Response: It is divided according to the calculated growth rate of total carbon emissions (Figure 3).
(30) In 5. Try to give a more in-depth explanation of your results. E.g. “ Although the coupling degree (average 0.93) between TCE, ED and EE in Lhasa is high, which is an important condition for the sustainable development of tourism, the average coupling coordination degree is only 0.63, which is at a primary coordinated level. There is still a lot of room for improvement.” Further elaboration on this is important. What do you want to say? Improvement in which direction?
Response: Thank you for your suggestions. It has further elaborated on the sentence change, and the future improvement direction is written in the follow-up policy recommendations.
(31) p. 13 Are the differences between the variation in carbon emission due to traffic, accommodation and activities statistically significant?
Response: Thank you for your suggestions. The total carbon emissions of transportation, accommodation and activities discussed in this paper are to explore the contribution of the three types of activities to the carbon emissions of tourism and their changes over the decade.
(32) In 5 & 6 you try sometimes to give explanations of some results without having evidence, neither from your analysis nor from the literature, please make sure to find some evidence for these statements. E.g. “ Results show that in 2014-2015, government policies accelerated the construction of tourism facilities, strengthened environmental protection, and rapidly increased environmental indicators, resulting in a decline in coupling. “. How do you know that it is the government policies that lead to these results? Did you test? Or this is your opinion? Do you have literature backing up your point of view? It would be very nice to have a statistically sound analysis that could help you not only to give a picture of the trends in all your indices but also to find out what could be the reasons for those changes. I would suggest the authors to create a cause-and-effect model to test some hypotheses related to the causes of the results obtained.
Response: Thank you for your suggestions. We have recognized this part, and added some references. The calculation results show that there was an inflection point in 2014-2015. According to the documents published by the government, the government introduced a series of policies to promote " green " and " environmental protection " during this period. The Lhasa municipal government also followed up a series of documents focusing on environmental protection, so it can be speculated that the government 's guidance led to this change.
(33) In 5&6 you present more the what you found rather than showing the implications of all this analysis. These parts should be backed up with literature review evidence, and the implications should be supported by additional literature that maybe wasn’t covered in the literature review but that could help support your discussion. In addition to this, it would be very important to clearly show what is your contribution to academia and the industry. This contribution is not very visible. I would like to suggest to the author to further develop this point to be capable to show more the contribution coming from this work.
(34) In general, tables should be reformatted. For example, it is not clear to which second-grade indices the third-grade indices belong to: add proper lines to distinguish
Response: Thank you for your suggestions. All similar table forms in the text have been modified.
(35) In Table 3 and 4 explain the phases and the sub-classes
Response: Thank you for your suggestions. C denotes the degree of coordination between the three subsystems. D denotes the coupling coordination degree. The definition of C and D has been explained in the previous article, and the secondary classification standard is based on the calculated value.
(36) Tell in 4. where the reader can find the results: i.e. Figure 3
Response: Thank you for your suggestions. As suggested, the results can be seen in the figure 6 have been added in the text.
(37) In the work, there is no limitation of the research and no future of the research. I would like to suggest to the authors to include this part to reflect on the process and identify opportunities for improvement but also to give scholars and idea for further development
Response: Thank you for your suggestions. The limitations of this paper and the prospect of future research have been increased.
Sincerely,
Authors

Reviewer 5 Report
Comments and Suggestions for Authors
The study addresses the significant issue of tourism-related greenhouse gas emissions, specifically highlighting vehicular traffic and accommodation as carbon dioxide sources (p.2). However, it is essential to emphasize that the carbon footprint of tourism extends beyond these factors, encompassing direct and indirect emissions from food and beverages, infrastructure development, waste management, and land use changes. Given the unique context of the Qinghai-Tibet Plateau, where the study is conducted, the impacts of these aspects hold extreme importance.
The evaluation of the interplay among economic growth, carbon emissions, and the ecological environment requires a comprehensive approach. While the study focuses on decoupling economic growth from carbon emissions, it is crucial to consider broader perspectives, including the transition to renewable energy, environmental quality, social equality, and overall ESG considerations on a long-term basis. Notably, the study misses important references related to permafrost responses, ecological research, and rangeland degradation on the Qinghai-Tibet Plateau, which could enhance the depth of the analysis (Cheng et al., 2007; Liu et al., 2021; Li et al., 2013).
The application of the Tapio model is acknowledged as a useful tool, but the study lacks a discussion on its limitations. While the model offers a valuable framework for understanding the decoupling process, it simplifies complex systems and assumes linear relationships, potentially neglecting the real-world intricacies, especially within the dynamic tourism sector. Notably, the model excludes external shocks, such as the impact of events like COVID, and does not adequately represent policy implementation challenges.
The study employs a coupling coordination degree model to analyze relationships between subsystems, but the simplicity of this model may oversimplify the nuances in the relationships between tourism carbon emissions, economic development, and the ecological environment. Furthermore, the definition of sustainability is not explicitly stated, and it is crucial to consider the social aspects neglected by both the coupling coordination degree model and the Tapio model.
The linear assumption for the period 2010-2020 oversimplifies the dynamic interactions between tourism, economic development, and the environment. A more nuanced understanding of these relationships, considering non-linear trends and external influences, would provide a more accurate representation of the complexities involved.
The study highlights the importance of government functions, technological innovation, and improving carbon productivity. However, it lacks an in-depth discussion on the feasibility and challenges of implementing these policy recommendations. This aspect is crucial for understanding the practical implications of the findings.
Feedback mechanisms regarding how changes in carbon emissions influence tourism activities are limited in the study. Understanding these feedback loops is essential for developing effective and adaptive policies that account for the dynamic nature of the tourism sector.
The stakeholder perspective is entirely omitted from the study, despite its critical role in tourism. Considering stakeholders, including local communities, tourists, and businesses, is vital for understanding the social dimensions of sustainability and aligning policies with the needs and values of those directly affected by tourism activities. Integrating stakeholder perspectives would enhance the study's applicability and provide a more comprehensive understanding of sustainable tourism development.
Author Response
Dear reviewer5,
Thank you for the careful review and valuable comments. We sincerely appreciate your suggestion and have carefully considered every opinion. In response to the comments, we have made substantial revisions to the article involving each section of the manuscript. In addition, Native English experts in ecology and economics were invited to polish the text. The specific responses are as follows:
(1) The study addresses the significant issue of tourism-related greenhouse gas emissions, specifically highlighting vehicular traffic and accommodation as carbon dioxide sources (p.2). However, it is essential to emphasize that the carbon footprint of tourism extends beyond these factors, encompassing direct and indirect emissions from food and beverages, infrastructure development, waste management, and land use changes. Given the unique context of the Qinghai-Tibet Plateau, where the study is conducted, the impacts of these aspects hold extreme importance.
Reply: Thanks very much for your suggestions. Definitely, food and beverages, infrastructure development, waste management, and land use changes are very important to analysis the tourism’s carbon footprint. However, the relative data is hard to obtain, especially in the Tibet. And this study focus on the relationship among the tourism’s carbon emissions, economy and ecological environment, so we choose factors vehicular traffic, accommodation and tourism events. The tourism event includes sightseeing, leisure, business, family visits, and other. We will study factors you mentioned in the future study.
(2) The evaluation of the interplay among economic growth, carbon emissions, and the ecological environment requires a comprehensive approach. While the study focuses on decoupling economic growth from carbon emissions, it is crucial to consider broader perspectives, including the transition to renewable energy, environmental quality, social equality, and overall ESG considerations on a long-term basis. Notably, the study misses important references related to permafrost responses, ecological research, and rangeland degradation on the Qinghai-Tibet Plateau, which could enhance the depth of the analysis (Cheng et al., 2007; Liu et al., 2021; Li et al., 2013).
Reply: Thanks very much for your suggestions. We added the broader perspectives and some important references in the introduction section according to your advice.
(3) The application of the Tapio model is acknowledged as a useful tool, but the study lacks a discussion on its limitations. While the model offers a valuable framework for understanding the decoupling process, it simplifies complex systems and assumes linear relationships, potentially neglecting the real-world intricacies, especially within the dynamic tourism sector. Notably, the model excludes external shocks, such as the impact of events like COVID, and does not adequately represent policy implementation challenges.
Reply: Thank you for your suggestions. We added the model’s limitation in the limitation section of the article according to your advice.
(4) The study employs a coupling coordination degree model to analyze relationships between subsystems, but the simplicity of this model may oversimplify the nuances in the relationships between tourism carbon emissions, economic development, and the ecological environment. Furthermore, the definition of sustainability is not explicitly stated, and it is crucial to consider the social aspects neglected by both the coupling coordination degree model and the Tapio model.
Reply: Thank you for your suggestions. We chose the coupling coordination degree model and the Tapio model, because them are useful widely used tool to analysis relationships between tourism carbon emissions, economic development, and the ecological environment. For their limitations, we added relative descriptions about oversimplify and neglecting the social aspects to the “Limitations and future work” section in the manuscript.
(5) The linear assumption for the period 2010-2020 oversimplifies the dynamic interactions between tourism, economic development, and the environment. A more nuanced understanding of these relationships, considering non-linear trends and external influences, would provide a more accurate representation of the complexities involved.
Reply: Thanks very much for your suggestions. This paper draws on the current widely used research models and ideas to analysis the relationships of subsystems. In the future study, we will study their nonlinear relationship in a deeper level. We added relative descriptions in the “Limitations and future work” section of the manuscript.
(6) The study highlights the importance of government functions, technological innovation, and improving carbon productivity. However, it lacks an in-depth discussion on the feasibility and challenges of implementing these policy recommendations. This aspect is crucial for understanding the practical implications of the findings.
Reply: Thanks very much for your suggestions. We added the relative description about the feasibility and challenges of implementing these policy recommendations in the “discussion” and “main policy recommendations” section.
(7) Feedback mechanisms regarding how changes in carbon emissions influence tourism activities are limited in the study. Understanding these feedback loops is essential for developing effective and adaptive policies that account for the dynamic nature of the tourism sector.
Reply: Thanks very much for your suggestions. We added the relative description in the “discussion” and “main policy recommendations” section according to your advice.
(8) The stakeholder perspective is entirely omitted from the study, despite its critical role in tourism. Considering stakeholders, including local communities, tourists, and businesses, is vital for understanding the social dimensions of sustainability and aligning policies with the needs and values of those directly affected by tourism activities. Integrating stakeholder perspectives would enhance the study's applicability and provide a more comprehensive understanding of sustainable tourism development.
Reply: Thanks very much for your suggestions. We added the relative description in the “discussion” and “main policy recommendations” section according to your advice

Reviewer 6 Report
Comments and Suggestions for Authors
The topic of the paper is interesting and actual, relevant for scientific research. Nevertheless, it is strongly recommended to improve the literature review at the beginning of the paper.
Author Response
Dear reviewer6,
Thank you for the careful review and valuable comments. We sincerely appreciate your suggestion and have carefully considered your opinion. In view of the comments, we have made substantial changes to the articles involved in the manuscript. The following is the specific response:
The topic of the paper is interesting and actual, relevant for scientific research. Nevertheless, it is strongly recommended to improve the literature review at the beginning of the paper.
Reply: Thanks very much for your suggestions. Some literature has been added to the literature review section based on your recommendations

Round 2
Reviewer 1 Report
Comments and Suggestions for Authors
- The manuscript exhibited a similarity level of 35%. The resemblance provided substantiation for the novelty and originality of the research. Consequently, 35% of the work was comparable to that of others. Twenty to twenty-five percent should be included in a manuscript fit for a journal of this caliber, such as Sustainability. A threshold of 20% similarity should be deemed unacceptable.
- The manuscript failed to adhere to the guidelines and template provided by Sustainability.
- Focus on clearly stating research gaps, objectives, and a more substantial justification for methodologies. These improvements improve the manuscript's clarity, relevance, and academic rigor. Specifically, define the research gap and objectives to contextualize the study in the field and show its necessity and relevance. It explains the research's unique contribution. Methodological Justification: A detailed justification for the methods ensures that the research approach is appropriate and rigorous, bolstering the findings' credibility and reliability.
- The data utilized in this study extended well beyond the present condition of the issue. The authors use the obsolete 2010-2020 dataset, which prevents them from illustrating the present state of affairs and needs to be more logical, particularly concerning tourism and the environment. The two-year-old data was satisfactory, albeit less intense than the previous year. The notions of travel and the environment have undergone significant transformations since 2020, encompassing aspects such as energy, waste, resource management, aesthetics, and cuisine.
- The manuscript appears to focus on mathematics rather than the field of Sustainability. There is no evidence available to indicate the source of the emission. China has recently been increasingly utilizing electric vehicles (EVs) for transportation, demonstrating a heightened focus on environmental issues.
- The dataset, discussions, suggestions, contributions, and innovative knowledge would likely become obsolete. There were numerous recurrent thoughts and conclusions. Figures 4, 6, and 7 all depicted the exact directions.

The manuscript's quality could be significantly improved. However, clarity and readability could be enhanced by improving grammatical accuracy, specifically in the usage of articles, punctuation, and commas, as well as by simplifying complex sentences. Improving the brevity of sentences and guaranteeing a coherent exposition of concepts will enhance the legibility of the manuscript for a wide range of readers. In order to optimize the general quality of the manuscript, it is imperative to conduct a comprehensive review for prevalent grammatical errors and eliminate run-on sentences. The initial sentence of the abstract was excessively lengthy and failed to provide essential motivation or comprehension. The English writing necessitated significant improvement and was polished professionally.
Author Response
Dear reviewer1,
Thank you for the careful review and valuable comments. We sincerely appreciate your suggestion and have carefully considered every opinion. In response to the comments, we have made substantial revisions to the article involving each section of the manuscript. In addition, Native English experts in ecology and economics were invited to polish the text. The specific responses are as follows:
(1) The manuscript exhibited a similarity level of 35%. The resemblance provided substantiation for the novelty and originality of the research. Consequently, 35% of the work was comparable to that of others. Twenty to twenty-five percent should be included in a manuscript fit for a journal of this caliber, such as Sustainability. A threshold of 20% similarity should be deemed unacceptable.
Reply: Thank you for your suggestions. We are very sorry for this. We checked the full text of the paper and found that this may be caused by our imitation of English sentence structure. Therefore, in the last revised manuscript, we revised the whole text by the repeated reports. Finally, the iThenticate anti-plagiarism system dedicated to SCI/EI/SSCI journals was used for self-examination, and the repetition rate was 18%.

(2) The manuscript failed to adhere to the guidelines and template provided by Sustainability.
Reply: Thank you for your suggestions. We downloaded the latest template and reorganized our manuscript following your guidelines.
(3) Focus on clearly stating research gaps, objectives, and a more substantial justification for methodologies. These improvements improve the manuscript's clarity, relevance, and academic rigor. Specifically, define the research gap and objectives to contextualize the study in the field and show its necessity and relevance. It explains the research's unique contribution. Methodological Justification: A detailed justification for the methods ensures that the research approach is appropriate and rigorous, bolstering the findings' credibility and reliability.
Reply: Thanks very much for your suggestions. We have organized and strengthened the parts you mentioned.
(4) The data utilized in this study extended well beyond the present condition of the issue. The authors use the obsolete 2010-2020 dataset, which prevents them from illustrating the present state of affairs and needs to be more logical, particularly concerning tourism and the environment. The two-year-old data was satisfactory, albeit less intense than the previous year. The notions of travel and the environment have undergone significant transformations since 2020, encompassing aspects such as energy, waste, resource management, aesthetics, and cuisine.
Reply: Thank you for your suggestions. We tried our best to find the new data, but can not find any relative data of Tibet from 2021 to 2023. This may due to the autonomouy of Tibet and its location on the southwest boundary of China. Data statistics lag behind those of other regions, especially affected by the pandemic. The data that can be fully collected is only up to 2020, so we chose the data from 2010 to 2020. However, we added possible deficiencies and limitations due to data reasons in the “limitations and future work” section.
(5) The manuscript appears to focus on mathematics rather than the field of Sustainability. There is no evidence available to indicate the source of the emission. China has recently been increasingly utilizing electric vehicles (EVs) for transportation, demonstrating a heightened focus on environmental issues.
Reply: Thank you for your suggestions. The aim of this paper is to analyze and reveal the coupling relationship between environment and tourism with mathematical methods. Although, there is no evidence available to indicate the source of the emission, numerous studies shown that the carbon emissions of tourism mainly come from the following aspects: transportation, accommodation and catering, operation of scenic spots and construction of tourist facilities. Tourist attractions on the Tibetan Plateau are almost natural landscapes, transportation and accommodation are major sources of emissions from tourism activities.
(6) The dataset, discussions, suggestions, contributions, and innovative knowledge would likely become obsolete. There were numerous recurrent thoughts and conclusions. Figures 4, 6, and 7 all depicted the exact directions.
Reply: Thanks very much for your suggestions. Because of the autonomouy of Tibet and its location on the southwest boundary of China. Data statistics lag behind those of other regions, especially affected by the pandemic. The data that can be fully collected is only up to 2020, so we can not add the new data from 2021 to 2023. However, we added possible deficiencies and limitations due to data reasons in the “limitations and future work” section. And we revised the results and discussion section, deleted figure 7 to make the text expression and structure more clear.

Reviewer 2 Report
Comments and Suggestions for Authors
Enclosed

Enclosed
Author Response
Dear reviewer2,
Thank you for the careful review and valuable comments. We sincerely appreciate your suggestion and have carefully considered every opinion. In response to the comments, we have made substantial revisions to the article involving each section of the manuscript. In addition, Native English experts in ecology and economics were invited to polish the text. The specific responses are as follows:
(1) “Environmental damage does not readily recover” Damages are not "recovered" but undone. Please have a native speaker review your article to make sure no Chinese English.
Reply: Thank you for your suggestions. According to your advice. It has been modified according to your opinions. And this manuscript was edited for proper English language, grammar, punctuation, spelling, and overall style by one or more of the highly qualified native English speaking editors at NativeEE. NativeEE specializes in editing and proofreading scientific manuscripts for submission to peer-reviewed journals.
(2) “sby” ?

Reply: Thank you for your suggestions. We are very sorry about that. It is a pen error, should be ' by ', has been modified in the article.
(3) “global” How to interpret global in this context?

Reply: Thank you for your suggestions. What this place wants to express is that the study of the relationship between human activities and the environment on the Qinghai-Tibet Plateau has attracted the attention of researchers around the world. To avoid ambiguity, we have adjusted the wording.
(4) “however, there has been less focus on the impacts of tourism has been negligibly examined” no two verbs in one sentence
![]()
Reply: Thank you for your suggestions. We are very sorry about that. It be modified here according to your opinion.
(5) “alien human” how to define "alien human"? Do you mean visitors who are not local to the environment? I would say "human activities from outsiders, not domiciled in the region".

Reply: Thank you for your suggestions. Here we want to express is not living in the local, from the outside of the population. We have modified them as “non-local human activities” and “local human activities”.
(6) “was” If this is what you are doing, why not present tense?

Reply: Thank you for your suggestions. It has been modified according to your opinion.
(7)“offer” please use gerund instead.

Reply: Thank you for your suggestions. This sentence has been reorganized according to your advice.
(8) “significant influence on the global climate,” How to justify its relative significance?

Reply: Thank you for your suggestions. When I read the literature, I found that the Qinghai-Tibet Plateau is the highest plateau in the world. Its glacier ablation, oxygen production, and thermal changes affect global ocean currents, sea surface temperature, monsoon and other climatic factors. Therefore, I express here that the role of the Qinghai-Tibet Plateau in the global climate is significant. Literature citations have been added there.
(9) “assumes” Have you formulated H1, H2, and H3 to articulate your hypothesis for further testing?

Reply: Thank you for your suggestions. What we want to express here is that because TCE, ED and EE are closely related to each other, but there is no direct evidence to explain their relationship with the sustainable development of tourism. Therefore, this paper assumes that these three factors are the key factors to promote the sustainable development of tourism. Through calculation and analysis, these three have indeed played an important role in the sustainable development of tourism.
(10) “Zhonghua Renmin Gongheguo Guowuyuan Gongbao” What does that mean in English? Does the State Department issued an English translation from which you may directly cite?

Reply: Thank you for your suggestions. We found the English translation and revised it according to your advice.
(11) “government departments should promote tourism by train and road vehicles and develop low-carbon fuels (biofuels or hybrid power sources) to decrease their carbon emissions.” Are there any specific policies you may cite?

Reply: Thank you for your suggestions. This is through the above analysis, because the carbon emissions generated by automobiles and railways are smaller than those of aircrafts, so from the perspective of environmental protection, it is hoped that the government can encourage people to choose a more environmentally friendly way to travel. On the other hand, due to the high carbon emissions generated by aircraft, it is recommended to develop more environmentally friendly fuels to reduce the carbon emissions of aircraft.
(12) “This paper is subject to the following limitations: (1) Taking into account the availability of data, we used data from 2010-2020. Future research could expand this interval to obtain further insights. (2) Lhasa was used as a representative city of plateau tourism. There are many beautiful plateau cities in the world. Future research could focus on multiple plateau cities and explore their similarities and differences.” Please make it into one paragraph, and address the methodological limitations of coupling of how to make it more general in future to cope with more diverse needs.

Reply: Thank you for your suggestions. This part has been combined into a paragraph. And how to use the coupling coordination model in the future is expounded, according to your advice.
(13) “In the planning and development of tourism activities” How about the legal aspect? What law can be promulgated and enforced to make the policies more down-to-earth?

Reply: Thank you for your suggestions. It is indeed a very important aspect to formulate legal rules to ensure the development of low-carbon tourism. In the policy recommendation section of the article, we have added suggestions to the relevant departments to formulate laws on energy conservation and emission reduction, and local governments to monitor the carbon emissions generated by enterprises in their own regions.
(14) “railway” In the previous sections, you mention both railroad and road, but why just one here?

Reply: Thank you for your suggestions. We are sorry about that the way of road is omitted here. We added it according to your advice.

Reviewer 3 Report
Comments and Suggestions for Authors
Dear authors,
Dear authors, you did a good job!
Author Response
Dear reviewer3,
Thank you very much for affirming us.
Reviewer 4 Report
Comments and Suggestions for Authors
Thank you very much for the improvements made to the work submitted. It is visible a hight commitment to it. While reading the modification it is possible to see a more clearer argument about the topic described and logic. I still see some limitations in actual contribution of the paper to academia. I understand well why it is relevant to understand some of these aspects on a practical level, but the academia and also the industry have already and understanding of the underlying factors identified and discussed in this paper. The originality of it looks limited and also the contribution to the literature is limited. It is interesting to see the evolution of the different measurements through the time and the possible link that exist between the policies implemented and the results obtained, however the cause-effect in these relationships should be more in depth explored. There is not an actual hypothesis testing that can tell us that those factors influence the results. You can suppose and probably this is correct, but maybe other factors for which you have to control could have influenced the relationship. Although the changes made the contribution of the work remains limited, if the goal is to explore much more in depth the relationships to bring a higher contribution to the academia and explore further aspects that might be already known in the literature. The novelty could be linked the plateau, but I am not convinced this is strong enough. I would suggest to improve this part. However in order to do so I would suggest additional analysis that show better the cause and effect relationship between results and factors that cause them. If this is not possible with the database available it would be at least recommended to suggest this analysis in the future of the research. Looking at the future of the research you mentioned that the same analysis could be carried out in other plateau. Why? What if instead you propose a future of the research that will take into consideration the results obtained in this research and further explore them and go in depth into other interesting mechanisms that might add more value to academia?
There are some small types, such as
- sby (first line p. 2)
- mail p.2 (13 lines from the bottom of the page)
I won’t use the terminology “hypothesis” in this paragraph “Hypothesis of sustainable tourism development” the word hypothesis. Have you tested hypothesis?
Be also careful with the word “correlation”: the correlation between carbon emissions induced by tourism and economic growth has been established (Akram, Yang & Hafeez, 2023; Tang, Shang, & Shi 2014) and the CCR of the tourism-economy-environment system has been confirmed (Xie et al., 2021; Pan, Weng & Li 2021).” Correlation doesn’t tell us what is the direction of the relationship, it is not a cause and effect relationship. It would be important to explain this concept further to let people understand what that is the relationship that was actually established. The more the economic growth the more carbon emission? Or the more carbon emission the more economic growth? I assume the first one, but if you don’t make this explicit it is not clear, for the reason that was explain before related to the correlation.
Overall the work is interesting as a starting point for further insights and researches that could make the paper more interesting and capable to elevate to a superior level. Critical analysis should be improved in the discussion of the conclusion and recommendations.
Author Response
Dear reviewer4,
Thank you for the careful review and valuable comments. We sincerely appreciate your suggestion and have carefully considered every opinion. In response to the comments, we have made substantial revisions to the article involving each section of the manuscript. In addition, Native English experts in ecology and economics were invited to polish the text. The specific responses are as follows:
(1) The novelty could be linked the plateau, but I am not convinced this is strong enough. I would suggest to improve this part. However, in order to do so I would suggest additional analysis that show better the cause and effect relationship between results and factors that cause them. If this is not possible with the database available it would be at least recommended to suggest this analysis in the future of the research.
Reply: Thanks for your suggestions. We tried to add 2021-2023 data for analysis, but because Tibet is an autonomous region in China, and the geographical location is relatively remote, the update of statistical data is relatively lagging behind, many data are not publicly released, we can not get all the data required for 2021-2023 to do more analysis. But we have written in this part of the limitations of this article that this article has certain limitations on the time scale. When the data is public, we can obtain updated data for analysis in future research.
(2) Looking at the future of the research you mentioned that the same analysis could be carried out in other plateau. Why? What if instead you propose a future of the research that will take into consideration the results obtained in this research and further explore them and go in depth into other interesting mechanisms that might add more value to academia?
Reply: Thanks very much for your suggestions. We revised this part in the “Limitations and future work” section according to your advice.
(3) sby (first line p. 2)
Reply: We are sorry about that is a pen error, should be ' by ', has been modified in the article.
(4) I won’t use the terminology “hypothesis” in this paragraph “Hypothesis of sustainable tourism development” the word hypothesis. Have you tested hypothesis?
Reply: Thanks very much for your suggestions. This paragraph is about the relationship between TCE, ED and EE, and the ' hypothesis ' has been changed to ' mechanism '.
(5) Be also careful with the word “correlation”: the correlation between carbon emissions induced by tourism and economic growth has been established (Akram, Yang & Hafeez, 2023; Tang, Shang, & Shi 2014) and the CCR of the tourism-economy-environment system has been confirmed (Xie et al., 2021; Pan, Weng & Li 2021).” Correlation doesn’t tell us what is the direction of the relationship, it is not a cause and effect relationship. It would be important to explain this concept further to let people understand what that is the relationship that was actually established. The more the economic growth the more carbon emission? Or the more carbon emission the more economic growth? I assume the first one, but if you don’t make this explicit it is not clear, for the reason that was explain before related to the correlation.
Reply: Thanks very much for your suggestions. We revised them according to your advice.
(6) Overall the work is interesting as a starting point for further insights and researches that could make the paper more interesting and capable to elevate to a superior level. Critical analysis should be improved in the discussion of the conclusion and recommendations.
Reply: Thanks very much for your suggestions. We revised the critical analysis in the discussion of the conclusion and recommendations according to your advice.

Round 3
Reviewer 1 Report
Comments and Suggestions for Authors
The manuscript's 37% similarity with references and 19% without raises concerns about the originality and novelty of the research. The manuscript exhibited a 7% similarity with the article titled "Coupling Coordination and Influencing Factors among Tourism Carbon Emission, Tourism Economics, and Tourism Innovation," published in MDPI, available at https://doi.org/10.3390/ijerph18041601.
A more thorough literature review might be needed to differentiate this work significantly from existing research. It is crucial for the manuscript to clearly articulate its unique contribution to the field, particularly regarding the equilibrium between tourism development and ecological protection in the Qinghai-Tibet Plateau.
The reviewer agrees with the research objective but points out that the findings and results are not adequately shown to support the objectives. This critique indicates that the manuscript may benefit from a more detailed presentation and discussion of the results, directly linking them to how they contribute to breakthrough research in sustainable tourism development.
The methodology section appears to lack robustness in terms of data validity and reliability testing. The use of mathematical models (TCE, ED, and EE calculations) requires a thorough validation process to ensure the accuracy and relevance of the data. The absence of correlation tests or explanations on how inaccuracies in data collection were addressed leaves room for doubt regarding the reliability of the findings. Moreover, the lack of a detailed explanation for the distribution of weights in Table 3 suggests that a clearer methodology section is needed to enhance the credibility of the research.
Comments on the Quality of English Language
Language usage issues, including grammar and punctuation errors, negatively impact the readability and professional quality of the manuscript. This is particularly problematic in the abstract, which is crucial for attracting the interest of potential readers. Investing in thorough proofreading and revision for clarity and correctness can significantly improve the manuscript's quality.
Author Response
Dear reviewer 1,
Thank you for the reviewers ' careful review and valuable comments. We sincerely appreciate your suggestion and have carefully considered every opinion. In response to the comments, we have made substantial revisions to the article involving each section of the manuscript. The specific responses are as follows:
(1)The manuscript's 37% similarity with references and 19% without raises concerns about the originality and novelty of the research. The manuscript exhibited a 7% similarity with the article titled "Coupling Coordination and Influencing Factors among Tourism Carbon Emission, Tourism Economics, and Tourism Innovation," published in MDPI, available at https://doi.org/10.3390/ijerph18041601.
Reply: Thank you for your suggestions. We are very sorry for this. We compared the full text of our paper with the paper "Coupling Coordination and Influencing Factors among Tourism Carbon Emission, Tourism Economics, and Tourism Innovation". Our paper aims to understand how tourism activities affect tourism sustainable development in Qinghai-Tibet Plateau, by analyzing the relationships among tourism, the environment and carbon emissions. We combined the Tapio decoupling model and the coupling coordination degree model to analyze the relationships. It can reveal more accurately the influence relationship and intensity of tourism behaviors from multiple dimensions. Therefore, we adjusted the structure of the paper and rewrote the abstract, introduction, and mechanism sections to more clearly state the contributions and innovations of our paper.
(2)A more thorough literature review might be needed to differentiate this work significantly from existing research. It is crucial for the manuscript to clearly articulate its unique contribution to the field, particularly regarding the equilibrium between tourism development and ecological protection in the Qinghai-Tibet Plateau.
Reply: Thanks for your suggestion. We have added the literature review with the ecological protection of the Qinghai-Tibet Plateau and the unique contribution of this article to this section.
(3)The reviewer agrees with the research objective but points out that the findings and results are not adequately shown to support the objectives. This critique indicates that the manuscript may benefit from a more detailed presentation and discussion of the results, directly linking them to how they contribute to breakthrough research in sustainable tourism development.
Reply: Thanks for your suggestion. We have increased the verification of the influence of TCE, EE and ED on the coupling coordination degree in the result part. And in accordance with your comments to modify the discussion of the results section.
(4)The methodology section appears to lack robustness in terms of data validity and reliability testing. The use of mathematical models (TCE, ED, and EE calculations) requires a thorough validation process to ensure the accuracy and relevance of the data. The absence of correlation tests or explanations on how inaccuracies in data collection were addressed leaves room for doubt regarding the reliability of the findings. Moreover, the lack of a detailed explanation for the distribution of weights in Table 3 suggests that a clearer methodology section is needed to enhance the credibility of the research.
Reply:Thanks for your suggestion. This paper tests the reliability and validity of the collected data. Because of the length of the article, the test results are not included in the article, and the explanation of the data passing the test of validity and reliability has been supplemented in the article. The index weight in Table 3 is confirmed by the entropy method, and the introduction of the entropy method has been supplemented.
(5)Language usage issues, including grammar and punctuation errors, negatively impact the readability and professional quality of the manuscript. This is particularly problematic in the abstract, which is crucial for attracting the interest of potential readers. Investing in thorough proofreading and revision for clarity and correctness can significantly improve the manuscript's quality.
Reply:Thank you for your suggestions. According to your advice, It has been modified according to your opinions. And this manuscript was edited for proper English language, grammar, punctuation, spelling, and overall style by one or more of the highly qualified native English speaking editors at NativeEE. NativeEE specializes in editing and proofreading scientific manuscripts for submission to peer-reviewed journals.

Reviewer 4 Report
Comments and Suggestions for Authors
Overall the work shows improvements and acknowledgement of some of the limits of this work have been identified. I still believe that the contribution to academia of this research could be improved. This requires additional research and collection of additional data, if not available, that could enable the authors to logically see the impacts of what discussed and the relations between the different variables. Overall I am happy with the effort that the authors shown in the improvements. I might suggest to consider a mix method to increase the contribution of the research. However I acknowledge that this might take time. But if you really want to make a difference this could be a good path for further development. I am wondering, what type of database you have? Could you think on the basis of it to use the data for statistical analysis? E.g. correlations, cause and effect analysis with regression or even SEM or other models? It would also interesting to add a bit more information on the type of data you have been able to collect. There are some information, and you shared also some limits, but this would be interesting.
Author Response
Dear reviewer 4,
Thank you for your careful review and valuable comments. We sincerely appreciate your suggestion and have carefully considered every opinion. In response to the comments, we have made substantial revisions to the article involving each section of the manuscript. The specific responses are as follows:
(1)Overall the work shows improvements and acknowledgement of some of the limits of this work have been identified. I still believe that the contribution to academia of this research could be improved. This requires additional research and collection of additional data, if not available, that could enable the authors to logically see the impacts of what discussed and the relations between the different variables. Overall I am happy with the effort that the authors shown in the improvements. I might suggest to consider a mix method to increase the contribution of the research. However I acknowledge that this might take time. But if you really want to make a difference this could be a good path for further development. I am wondering, what type of database you have? Could you think on the basis of it to use the data for statistical analysis? E.g. correlations, cause and effect analysis with regression or even SEM or other models? It would also interesting to add a bit more information on the type of data you have been able to collect. There are some information, and you shared also some limits, but this would be interesting.
Reply: Thanks for your suggestion. We selected seven indicators from the three aspects of TCE, EE and ED, and analyzed the relationship between them and the degree of coupling coordination with geographic detectors. It is verified that there is an inseparable relationship between TCE, EE and ED and coupling coordination degree.
Best wishes,
Authors

Round 4
Reviewer 1 Report
Comments and Suggestions for Authors
The authors have adequately revised the manuscript to adhere to the quality and standards of the journal. However, a native English speaker needs to proofread the English usage. Most of the sentences were laborious and complex. It is advisable to present the complete designation of an abbreviation only once. The given abbreviation did not explicitly state the complete full name. The manuscript was difficult to understand and required greater reader engagement and excitement.
The manuscript exhibited a reduction in the degree of similarity at the accepted boundary.
Comments on the Quality of English LanguageA native English speaker needed to proofread the English usage. Most of the sentences were laborious and complex. It is advisable to present the complete designation of an abbreviation only once. The given abbreviation did not explicitly state the complete full name. The manuscript was difficult to understand and required greater reader engagement and excitement.
Author Response
Dear reviewer 1,
Thank you for your careful review and valuable comments. We sincerely appreciate your suggestion and have carefully considered every opinion. In response to the comments, we have made substantial revisions to the article. The specific responses are as follows:
(1)The authors have adequately revised the manuscript to adhere to the quality and standards of the journal. However, a native English speaker needs to proofread the English usage. Most of the sentences were laborious and complex. It is advisable to present the complete designation of an abbreviation only once. The given abbreviation did not explicitly state the complete full name. The manuscript was difficult to understand and required greater reader engagement and excitement.
Reply: Thanks for your suggestion. The abbreviations of the full text have been checked and modified.
(2) The manuscript exhibited a reduction in the degree of similarity at the accepted boundary.
Reply: Thanks for your suggestion. We have rechecked and modified again.
(3) A native English speaker needed to proofread the English usage. Most of the sentences were laborious and complex. It is advisable to present the complete designation of an abbreviation only once. The given abbreviation did not explicitly state the complete full name. The manuscript was difficult to understand and required greater reader engagement and excitement.
Reply: Thanks for your suggestion. According to your advice, this manuscript was edited for proper English language, grammar, punctuation, spelling, and overall style by one or more of the highly qualified native English speaking editors at NativeEE. NativeEE specializes in editing and proofreading scientific manuscripts for submission to peer-reviewed journals.
Best wishes,
Authors

Reviewer 4 Report
Comments and Suggestions for Authors
Dear Authors,
Thank you very much for resubmitting the paper. I really appreciate the attempt to show more the cause and effect that exists between TCE, ED, ED and D.
Here are some comments to further improve the section titled "Index Verification"
1) I would give a more effective title
2) Very good to show the relationship between TCE, EE, ED and D but inform the reader why do you think this is important to show in the discussion. Why is it relevant to do this analysis: A11, A21, A24, B1, B13, and C11, C22 and used geographic detectors to verify their impact on D
3) Could you please explain why you only selected 7 factors (A11, A21, A24, B1, B13, and C11, C22) within TCE, EE and ED?
4) what about the all the other factors? Did you exclude them because they are not significant? I would add in the table also the significance of all the others although not significant: these might be interesting findings as well.
5) I know there might be a limit in words but a reader that sees the A11, A21, etc. and doesn't remember what they mean won't be happy to go and read the meaning of them in the table 2, and go back and forth. Please add their meaning in table 6 or in text
6) what is the methodology you have used to carry out this analysis? Explain it
Other comments:
a) Leave some space before 4.5. Coupling relationship between subsystems in different stages
b) In several parts you speak about variation between years, e.g., "Tapio index had a wide range of variation" but is this variation statistically significant? Do you have test results? Show them
Overall I see improvements a more structured in this work.
Author Response
Dear reviewer 4,
Thank you for your careful review and valuable comments. We sincerely appreciate your suggestion and have carefully considered every opinion. In response to the comments, we have made substantial revisions to the article. The specific responses are as follows:
(1)Here are some comments to further improve the section titled "Index Verification". I would give a more effective title
Reply: Thanks for your suggestion. The subheading name has been changed.
(2)Very good to show the relationship between TCE, EE, ED and D but inform the reader why do you think this is important to show in the discussion. Why is it relevant to do this analysis: A11, A21, A24, B1, B13, and C11, C22 and used geographic detectors to verify their impact on D
Reply: Thanks for your suggestion. The reasons for this analysis have been explained in the article.
(3)Could you please explain why you only selected 7 factors (A11, A21, A24, B1, B13, and C11, C22) within TCE, EE and ED?
Reply: Thanks for your suggestion. The reasons for the selection of these 7 factors have been explained in the article.
(4)what about the all the other factors? Did you exclude them because they are not significant? I would add in the table also the significance of all the others although not significant: these might be interesting findings as well.
Reply: Thanks for your suggestion. The calculation results of all indicators have been added to Table 6.
(5)I know there might be a limit in words but a reader that sees the A11, A21, etc. and doesn't remember what they mean won't be happy to go and read the meaning of them in the table 2, and go back and forth. Please add their meaning in table 6 or in text
Reply: Thanks for your suggestion. The meaning of indices has been added to table 6.
(6) what is the methodology you have used to carry out this analysis? Explain it
Reply: Thanks for your suggestion. We explained the method principle of the geographical detector at 3.4.3.
(7)Leave some space before 4.5. Coupling relationship between subsystems in different stages
Reply: Thanks for your suggestion. It has been modified according to your suggestion.
(8) In several parts you speak about variation between years, e.g., "Tapio index had a wide range of variation" but is this variation statistically significant? Do you have test results? Show them
Reply: Thanks for your suggestion. The change in the Tapio index is shown by calculation in Figure 7. This place was originally intended to describe the overall trend of the Tapio index over the study period. In order to pave the way for the follow-up study in stages according to the specific change trend of Tapio index. We find it is not accurate to use numerical values to represent the trend of change, and the expression here has been modified in the text.
Best wishes,
Authors
